# Zero shot molecular generation via similarity kernels

Rokas Elijošius ®[1] ✉, Fabian Zills ®[2], Ilyes Batatia[1], Sam Walton Norwood[3], Dávid Péter Kovács ®[1], Christian Holm ®[2] & Gábor Csányi ®[1]

Generative modelling aims to accelerate the discovery of novel chemicals by directly proposing structures with desirable properties. Recently, score-based, or diffusion, generative models have significantly outperformed previous approaches. Key to their success is the close relationship between the score and physical force, allowing the use of powerful equivariant neural networks. However, the behaviour of the learnt score is not yet well understood. Here, we analyse the score by training an energy-based diffusion model for molecular generation. We find that during the generation the score resembles a restorative potential initially and a quantum-mechanical force at the end, exhibiting special properties in between that enable the building of large molecules. Building upon these insights, we present Similarity-based Molecular Generation (SiMGen), a new zero-shot molecular generation method. SiMGen combines a time-dependent similarity kernel with local many-body descriptors to generate molecules without any further training. Our approach allows shape control via point cloud priors. Importantly, it can also act as guidance for existing trained models, enabling fragment-biased generation. We also release an interactive web tool, ZnDraw, for online SiMGen generation (https://zndraw.icp.uni-stuttgart.de).

The combinatorial scaling of the available chemical space with molecule size is one of the main challenges in the design of new molecules and materials. Generative modelling aims to solve this by directly proposing structures with desirable properties, without exhaustively enumerating and screening the candidates. Recently, diffusion-based models have achieved impressive results in molecular docking[1] and generation of linkers[2], drug-like molecules[3,4] and crystal structures[5,6].

Diffusion models are trained to reverse a stochastic noising process, which gradually corrupts samples of training data until they are indistinguishable from samples drawn from an uninformative prior distribution, such as a standard Gaussian[7–9]. Once the reverse of this process is learnt, it can be applied to transform easily-sampled random noise into independent samples from the approximate data distribution. In particular, diffusion models regress the score $s(\boldsymbol{x}, t)$, defined as the gradient of the log-likelihood of a time-dependent distribution $p(\boldsymbol{x}; t)$ that interpolates between the data distribution at $t = 0$,

$p(\boldsymbol{x}; 0) = p_{\text{data}}(\boldsymbol{x})$, and a Gaussian distribution at $t = T$, $p(\boldsymbol{x}; T) = N(\boldsymbol{x}; \boldsymbol{\mu} = \boldsymbol{0}, \boldsymbol{\Sigma} = \boldsymbol{I}\sigma(T))$.

$$s(\boldsymbol{x}, t) = \nabla_{\boldsymbol{x}} \log p(\boldsymbol{x}; t) \tag{1}$$

Using the score, we can generate new samples by numerically integrating a stochastic differential equation describing the time reversal of the noising process, which has a known analytic form[9]. Because the data distribution is provided only in the form of samples, the score is intractable analytically. However, it can be efficiently learnt via an implicit objective that has a closed form when the limiting distribution at $t = T$ is taken to be Gaussian, an approach known as denoising score matching[10–12].

In the context of molecule generation, the score is closely related to atomic forces. Consider training data that comprise configurations sampled using molecular dynamics or other methods from an

[1]Engineering Laboratory, University of Cambridge, Cambridge, UK. [2]Institute for Computational Physics, University of Stuttgart, Stuttgart, Germany. [3]Department of Energy Conversion and Storage, Technical University of Denmark, Kgs. Lyngby, Denmark. ✉e-mail: re344@cam.ac.uk

underlying Boltzmann distribution, $x \sim \exp(-\beta U(x))/Z$. Here, $x = \{r, z\}$ is a set that represents a molecule, with $r$ the atomic positions and $z$ the chemical elements, $U(x)$ the potential energy, $\beta$ the inverse temperature, and $Z$ the partition function. In this case, when the elements $z$ are fixed, the score of the data distribution $s(x, 0)$ corresponds to the atomic force (defined as the negative gradient of the potential energy) up to a multiplicative constant:

$$s(x, 0) = \nabla_r \log p(x; t = 0) = -\beta \nabla_r U(x) = \beta F(x) \quad (2)$$

The significance is that the atomistic modelling community has built up an extensive body of knowledge on building excellent models of atomic forces in the form of force fields[13–20], lessons from building machine learning force fields[21–24] being particularly relevant. Several groups have already used this score-force relationship to pre-train networks for molecular property prediction[25], to improve the quality of generated structures[5,26], or to obtain coarse-grained force fields[27].

Starting from a random configuration and using a force-like quantity to move atoms and generate new structures closely resembles ab initio random structure search (AIRSS), pioneered by Pickard and Needs[28] for crystal structures. In AIRSS, atoms are initialised with random positions and then their energy or enthalpy minimised (relaxed, in common parlance) using quantum mechanical (QM) forces, i.e., $s(x, t) = F_{QM}(x)$. Despite its simplicity, AIRSS has achieved great success in discovering stable structures of periodic materials[29,30], especially at high pressures. However, it is not directly suitable for generating complex molecules. Applying AIRSS for molecular generation samples the QM energy surface, in other words, the Boltzmann distribution. This turns out to be very uninteresting: The probability density for molecular systems is overwhelmingly concentrated on the most stable chemical compounds, which are simple molecules such as water, carbon dioxide, and dinitrogen. Generative modelling requires a way to guide the generation towards molecules of interest, and by replacing the learnt score with analytical QM forces, that would be lost. Instead, we need a method that allows such directed generation, but at the same time leverages the information represented by the available analytical atomic forces.

A further problem of using AIRSS for molecule generation is that moving atoms along atomic forces does not allow for a change in composition. This is not a good restriction to have for generating large molecules, because local composition is closely related to local geometry. To retain compositional degrees of freedom in the generative process, score-based models split the score into two components: a positional component $s_r(x, t)$ that resembles the physical force and an elemental component $s_z(x, t)$, best interpreted as an alchemical force,

$$s(x, t) = \underbrace{\frac{\delta}{\delta r} \log p(x; t) dr}_{s_r(x, t)} + \underbrace{\frac{\delta}{\delta z} \log p(x; t) dz}_{s_z(x, t)} \quad (3)$$

The score of the positions $s_r(x, t)$ obeys the same symmetries as a physical force; namely, it must be translationally invariant and rotationally equivariant. On the other hand, $s_z(x, t)$ should be both translationally and rotationally invariant.

Given the symmetry requirements and the success of equivariant graph neural networks in QM force prediction[31–33] it is no surprise that they have been widely adopted for learning the score in molecular generative models[34–37]. Locality is a key principle in machine learning force fields, typically imposed by only allowing atoms that are within a certain cutoff distance of each other to communicate. This enables scaling to large structures by providing linear computational complexity in the number of atoms, and captures the physical intuition that most interactions are short-ranged. However, current generative models for molecules have typically employed global models which represent molecules as fully-connected graphs. While global models

have so far shown impressive results, they may face potential bottlenecks when scaling to very large systems. Furthermore, they may not fully exploit the inductive bias of locality when learning the force-like positional score $s_r(x, t)$.

Score-based generative models are usually trained to output the score directly, e.g., by applying a linear projection to the learnt features. An alternative is to enforce the conservation of probability by training an energy-based diffusion model[38,39], where the score is fitted as a derivative of the model's output. The energy-based formulation comes with a higher computational cost, requiring gradient back-propagation through the network to make predictions. However, the integral of the score corresponds to a learnt molecular potential energy and enables us to study the "energy landscape" of the trained generative model. We find in the following that this surface is surprisingly smooth and inherently penalises fragmentation. We suggest that these properties of the learnt energy together with the alchemical force $s_z(x, t)$ enable diffusion-based models to build larger molecules, compared to the conceptually related AIRSS method, which often generates fragmented molecules.

To further explore the design space of score-based generation, particularly the role of locality, we introduce Similarity-based Molecular Generation (SiMGen). This zero-shot method leverages a time-varying local similarity kernel and pretrained MACE descriptors to construct a smooth log-likelihood landscape. The MACE descriptors are locally defined; thus, the entire generation is also local and extensive, allowing us to construct molecules of arbitrary size, as well as perform conditional generation with strictly no changes to the model. Beyond zero-shot generation, we demonstrate SiMGen's crucial ability to act as guidance for existing trained models, enabling fragment-biased generation, a key application in drug discovery. Control of both generation and guidance is achieved simply by substituting a new reference set of local environments into the similarity kernel. We also explore how SiMGen can be combined with arbitrary spatial priors, allowing for precise control over the final shape of the generated structure, which we use to build a macrocycle-like molecule with over 100 heavy atoms. Finally, we release an interactive web browser based tool that enables users to easily carry out conditional generation, which we showcase by building a linker between two molecular fragments.

## Results and discussion
### Energy landscape of a diffusion model
We start our preliminary investigation by training an energy-based diffusion model, and assume that the time-dependent probability connecting the data distribution to a Gaussian takes the form

$$p(x; t) = \exp(-E(x; t))/Z(t). \quad (4)$$

With this formulation, the score is the derivative of a learnt time-dependent energy $E(x; t)$.

$$s_r(x, t) = -\nabla_r E(x, t), \quad (5)$$

$$s_z(x, t) = -\nabla_z E(x, t). \quad (6)$$

Here, $E(x; t)$ is parametrised using a MACE model, trained using denoising score matching[10–12].

Linear interpolation in time between a quadratic potential and the QM energy provides a natural baseline to compare the behaviour of the learnt generation process to:

$$E_{linear}(x; t) = \left(1 - \frac{t}{T}\right) E_{QM}(x) + \frac{t}{T} ||r||_2^2 / \left(2\sigma(T)^2\right) \quad (7)$$

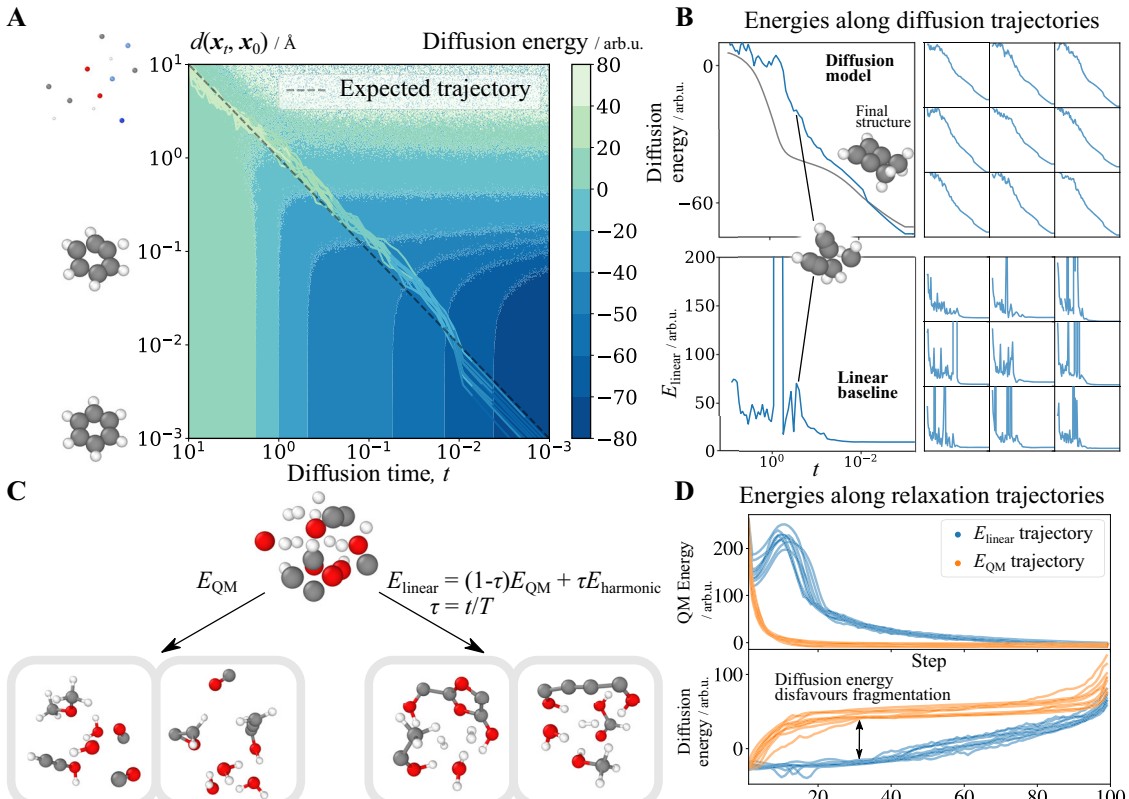

**Fig. 1 | Learnt energy landscape of a diffusion model is smooth and penalises fragmentation. A** The energy-based diffusion model learns a smooth energy landscape. The landscape depicts the learnt energy of a benzene molecule at varying deformation levels (quantified by $d(x_t, x_0)$, the measure of noise between a configuration $x_t$ and the reference structure $x_0$; see main text) and diffusion time ($t$, unitless). Coloured lines represent real generation trajectories, their colours correspond directly to the diffusion energy scale (see colour bar). While these trajectories do not always yield benzene, their energy profiles follow similar patterns. **B** Energy profiles of diffusion trajectories. Across ten examples, the learnt energy smoothly deceases during generation, even when the atoms overlap. In contrast, $E_{linear}$ evaluated on the same trajectories has a much rougher profile. The grey line indicates an ordinary differential equation (ODE) solver trajectory, and blue lines represent stochastic differential equation (SDE) solver trajectories. Insets illustrate molecular configurations at selected points. **C** Baseline methods using just QM force fields generate fragmented molecules. Relaxing using quantum mechanical (QM) forces, i.e., doing AIRSS exactly, leads to small, fragmented molecules. Using a linear baseline which mixes QM energy ($E_{QM}$) with a harmonic restorative potential ($E_{harmonic}$) results in larger but high-energy molecules. $\tau$ is a unitless generation coordinate, which progresses from $\tau = 1$ at the start to $\tau = 0$ at the end of a generation. AIRSS = ab initio random structure search. **D** Diffusion models penalise fragmentation. Evaluating diffusion energies along relaxation trajectories demonstrates a learnt penalty against molecular fragmentation. For $E_{linear}$ relaxation trajectories, the QM energy initially increases as the potential is dominated by the restorative term.

This linear baseline is loosely analogous to AIRSS with an initially very high pressure, where forces are assumed to be dominated by the confinement, with the pressure decreasing during generation. Like AIRSS, it requires a fixed composition and can only sample the global Boltzmann distribution. The time intervals where the learnt energy, $E(x, t)$, differs most from this linear baseline are thus the most critical. In these regions, the diffusion model has learnt to guide generation towards chemical space similar to the training data, rather than towards the modes of the global Boltzmann distribution.

The learnt energy depends on two variables: the time $t$, and the configuration $x$. We probe its behaviour by constructing a two-dimensional landscape over $t$ and a collective variable $d(x_t, x_0)$. Specifically, $d(x_t, x_0)$ is the standard deviation of atomic distances between the positions at time t, $x_t$, and the positions of the finally generated molecule, $x_0$.

$$d(x_t, x_0) = \mathrm{std}(\{|r_{1,t} - r_{1,0}|, \ldots, |r_{N,t} - r_{N,0}|\}) \qquad (8)$$

We call $d$ the "deformation level" of $x_t$ with respect to $x_0$. The construction of samples with a specific deformation level is easy and mimics the way samples are constructed during training (see Section III B). Each point in the landscape corresponds to a time $t$ and configuration $x_t$ with a specific deformation $d(x_t, x_0)$.

Panel A in Fig. 1 shows this landscape using benzene as $x_0$. Two striking features emerge. First is the clear dependence on time. At the start of generation ($t > 10^0$), the energy is not sensitive to deformation levels, increasing only for the most distorted configurations. However, as $t \to 0$, the energy increases steeply away from the equilibrium geometry. This shows that the model has learnt to first act as a simple restorative potential before becoming more chemical near the end. Second is the surprising smoothness of the landscape, best seen in Panel B (top), showing the energy along a real diffusion trajectory. The learnt energy smoothly decreases as the "soup of atoms" becomes a molecule (the small energy fluctuations are the result of using a stochastic sampler, not part of the landscape; a deterministic trajectory, grey line, is completely smooth). Compared to $E_{linear}$ evaluated on the same trajectory (Panel B bottom), the largest differences appear at intermediate $t$ ($10^0 < t < 10^{-1}$). Here, atoms often overlap or have unusual valences, causing the QM energy to diverge. In contrast, the diffusion model can handle such irregular configurations while the energy smoothly decreases throughout the generation.

More importantly, the diffusion model has learnt to penalise fragmentation. To show this, we generate molecules by minimising $E_{QM}$ or $E_{linear}$ of randomly initialised configurations. In both cases, we first remove any overlaps by relaxing with a Morse potential[40]. Panel C shows that the resulting structures are fragmented, with water and

hydrogen being particularly frequent "side products". The high bond dissociation energies make the formation of these small molecules irreversible, forcing the remaining atoms into extremely unsaturated and high-energy structures. This highlights the importance of the alchemical force $s_z(x; t)$: It allows diffusion models to dynamically change the element composition, preventing extremely unsaturated structures and enabling changes to the molecular graph without breaking bonds.

Panel D shows how the QM and diffusion model's energies change over the course of relaxation. Minimising $E_{\text{linear}}$ causes the QM energy to initially increase due to the restorative part of the potential. In addition, it produces larger molecules as the high pressure keeps the atoms together long enough for them to form bonds. The diffusion model consistently assigns a higher energy to the $E_{\text{QM}}$ relaxation trajectories, showing a bias towards larger, i.e., less fragmented, molecules.

Máté and Fleuret[41] recently investigated different interpolation schemes between prior and target densities for flow-based generative models. They showed that a trainable middle potential is needed to ensure coverage of the different modes in the target density. Our results indicate that, for molecular generation, energy-based diffusion models learn the key aspects of such a specialised potential without explicit separation of $E(x; t)$ into distinct components during training. Key features of this middle potential include smoothness despite distorted configurations, element swapping with the alchemical force, and bias toward training data-like structures.

## Zero shot generation with similarity kernels and evolutionary algorithms

Diffusion models have established themselves as a powerful approach for 3D molecular generation. While ongoing research refines diffusion-based methods[37], their design space remains vast, and exploring alternative approaches can lead to insights that advance the whole field. Building upon the analytical priors discussed previously, we introduce Similarity-based Molecular Generation (SiMGen), a zero-shot approach for 3D molecular generation. SiMGen leverages a similarity kernel to maximise local atomic similarity and an evolutionary algorithm to optimise elemental composition, enabling effective sampling from a reference distribution without task-specific training.

Focusing on local atomic environments allows SiMGen to explore distinct aspects of score-based generative models, offering a complementary perspective to global diffusion-based methods. Specifically, SiMGen allows us to examine:

- **Necessity of global architectures:** Diffusion models usually employ global architectures where the score function considers the entire structure. While effective, the computational cost can be significant for large systems. With SiMGen, we investigate whether a global architecture is fundamentally necessary for effective molecular generation, and if so, to what extent.
- **Conditioning and constraint incorporation:** While diffusion models offer conditioning via training or energy-based guidance, incorporating specific structural or similarity-based constraints can be complex. SiMGen offers a more direct approach to incorporating such constraints, and its ability to guide trained models highlights this advantage, providing a simpler route to controlled generation.
- **Data efficiency:** Training diffusion models requires substantial datasets, often tens of thousands of molecules. While transfer learning can reduce this, adapting to new domains can remain compute-intensive. SiMGen, leveraging a pre-trained representation, demonstrates data efficiency, requiring as few as tens of reference molecules to operate.

More broadly, SiMGen represents a design philosophy within score-based modelling that emphasises locality and the decoupling of representation learning from task-specific training. We begin with a discussion of how to use similarity kernels as generative models.

**Similarity kernels as generative models.** Previously, we used $x = \{r, z\}$ to represent whole molecules. We can also represent a molecule as a collection of atomic environments $x = \{\chi_1, \ldots, \chi_N\}$, where $\chi_i$ is a descriptor vector, representing the local environment around the atom $i$. The kernel function $k(\chi_i, \chi_j)$ is a measure of similarity between $\chi_i$ and $\chi_j$. By maximising this measure, we can directly steer the generation towards a chemical space defined by a set of reference atomic environments $\mathcal{D}_{\text{ref}} = \{\chi_1, \ldots, \chi_N\}$.

The correct choice of kernel and its parameters is essential for this strategy to work. Here we use a time-dependent version of the radial basis function (RBF) kernel:

$$k(\chi_i, \chi_j; t) = \exp\left(-\frac{||\chi_i - \chi_j||^2}{2\sigma(t)^2}\right), \tag{9}$$

The RBF kernel is a universal approximator but more importantly for us the kernel width $\sigma(t)$ controls the locality of the kernel and its gradient[42]. When $\sigma(t)$ is small, the kernel and its gradient rapidly decay to zero as $||\chi_i - \chi_j||$ increases, while when $\sigma(t)$ is large, the decay is much more gradual. This ability to control the locality of the kernel is essential, as we discuss further on.

Since we want the kernel to measure local similarity, we also need the representations $\chi_i$ to be local. While any local scheme, such as ACE[24] or SOAP[43], could be used to generate $\chi$, we found that learnt descriptors from a pretrained machine learning (ML) potential perform better. Section III A 1 gives details on the construction of such descriptors.

To use the kernel as a generative model, we construct an energy function defined by the reference environments $\mathcal{D}_{\text{ref}}$. We define the similarity energy of an atom as the negative log likelihood of its environment with respect to the reference data.

$$E_{\text{sim}, i} = -\log f(\chi_i, t) = -\log \sum_{j \in \mathcal{D}_{\text{ref}}} k(\chi_i, \chi_j; t) \tag{10}$$

Summing over the atoms yields the total similarity energy of a configuration.

$$E_{\text{sim}}(x; t) = \sum_{i \in x} E_{\text{sim}, i} = -\sum_{i \in x} \log\left(\sum_{j \in \mathcal{D}_{\text{ref}}} k(\chi_i, \chi_j; t)\right) \tag{11}$$

$$F_{\text{sim}}(x; t) = -\nabla_x E_{\text{sim}}(x; t) = \nabla_x \sum_{i \in x} \log f(\chi_i; t). \tag{12}$$

$E_{\text{sim}}(x; t)$ quantifies the similarity of atomic environments in $x$ to the reference set $\mathcal{D}_{\text{ref}}$ – the more similar the environments, the lower the energy. The force on an individual atom can be understood as the direction that maximises the local similarity to the reference data. This similarity force generalises the work of Cobelli et al.[44], allowing the generation of environments that are similar but distinct from the reference data. Crucially, since we are using local descriptors, the approach is independent of the global size of the molecules from which these environments are taken. The descriptors for each atom are mainly determined by its closest neighbours, meaning it makes little difference whether the environments come from a large molecule or a small one. This contrasts globally connected generative models, where the size of the underlying molecular graph can represent a real distributional shift for the model during training and inference.

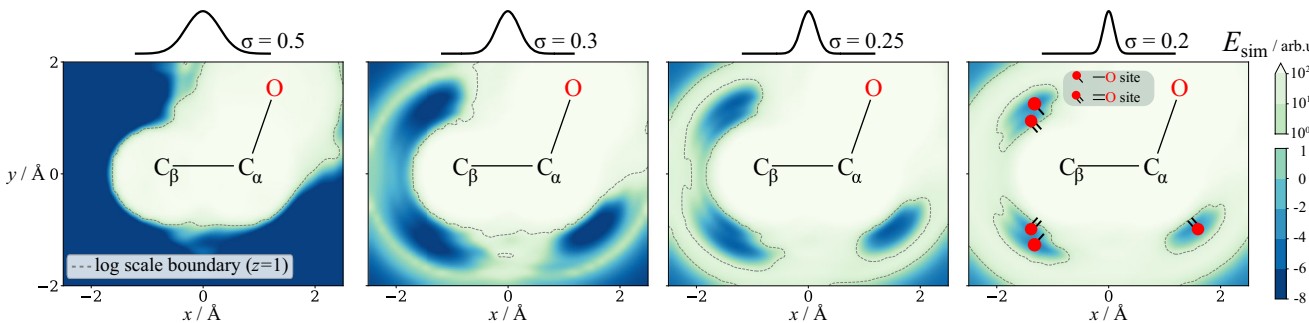

**Fig. 2 | The kernel width modulates the similarity energy landscape.** We probe the similarity energy, $E_{sim}$, by placing an ethanol molecule without hydrogens in the $xy$ plane and measuring the value of $E_{sim}$ if an additional oxygen atom is placed at each $x$, $y$ position. Varying the width of the similarity kernel, $\sigma(t)$, creates a funnelling effect that gradually guides atoms from the relatively flat energy surface at large $\sigma$ values towards specific minima as $\sigma$ decreases.

In state-of-the-art diffusion models, the score function is constructed in such a way that it always points in the direction of the data[45,46]. We can emulate this effect by setting a large initial kernel width $\sigma(t)$ and then reducing it as the generation progresses. Expanding Equation (12) shows that the similarity force consists of weighted contributions from all reference environments.

$$F_{sim}(\boldsymbol{x}; t) = \frac{1}{\sigma(t)^2} \sum_{i \in \boldsymbol{x}} \sum_{j \in \mathcal{D}_{ref}} w_j (\boldsymbol{\chi}_j - \boldsymbol{\chi}_i)^T \nabla_r \boldsymbol{\chi}_i \tag{13a}$$

$$w_j = \frac{\exp\left(-\|\boldsymbol{\chi}_i - \boldsymbol{\chi}_j\|^2 / (2\sigma(t)^2)\right)}{\sum_{j \in \mathcal{D}_{ref}} \exp\left(-\|\boldsymbol{\chi}_i - \boldsymbol{\chi}_j\|^2 / (2\sigma(t)^2)\right)} \tag{13b}$$

$$= \text{softmax}(-\|\boldsymbol{\chi}_i - \boldsymbol{\chi}_j\|^2 / (2\sigma(t)^2)) \tag{13c}$$

When $\sigma(t)$ is large, all reference environments contribute equally per the softmax, and $F_{sim}(\boldsymbol{z}; t)$ points towards the "mean atomic environment". As $\sigma(t) \to 0$, only the closest reference environment dominates the force acting on each atom. This transition from large to small $\sigma(t)$ enables atoms to first explore possible arrangements before ultimately settling into specific local minima.

We demonstrate how the width of the kernel affects the similarity energy in Fig. 2. Each individual plot is constructed by placing an ethanol molecule without hydrogen atoms in the XY plane. The energy at each $(x, y)$ corresponds to $E_{sim}$ if an additional oxygen atom is placed at that coordinate. At the largest $\sigma$ the repulsive terms dominate, as the "mean atomic environment" does not contain environments where the atoms overlap. As we decrease $\sigma$, specific minima emerge. Next to the $\beta$-carbon, 4 minima appear. These minima correspond to the formation of either a single bond or a double bond to the new oxygen atom. Only one minimum emerges next to the $\alpha$-carbon. The O–C–O fragment can be planar only if one of the oxygen atoms is double-bonded to the carbon, making an O = site the only possibility. Varying $\sigma(t)$ is crucial for the generation process, as no single value of $\sigma(t)$ is appropriate for the whole generation.

The core of our method is the similarity energy and force $E_{sim}$ and $F_{sim}$. However, to match the correct distributions at the two endpoints, we need to combine the similarity force with a quadratic restorative force early in the generation and the actual quantum mechanically derived force near the end.

$$\begin{aligned} F(\boldsymbol{x}; t) = {} & k_{prior}(t) F_{prior}(\boldsymbol{x}) + k_{sim}(t) \tilde{F}_{sim}(\boldsymbol{x}; t) \\ & + k_{QM}(t) F_{QM}(\boldsymbol{x}) \end{aligned} \tag{14}$$

$F_{prior}(x)$ is the gradient of the log likelihood of the prior distribution. In the case of standard Gaussian prior $F_{prior}(x) = -\boldsymbol{r}$. In theory, scheduling

functions $k_*$ are constrained such that $F(\boldsymbol{x}; T) = F_{prior}(\boldsymbol{x})$ and $F(\boldsymbol{x}; 0) = F_{QM}(\boldsymbol{x})$. In practice, we use the similarity force from the start, as moves guided solely by the uninformative prior waste computational effort without contributing to the final result. Equation (13a) suggests a natural schedule, $k_{sim}(t) = 1/\sigma(t)^2$, for the similarity force, and we thus rewrite similarity force as the product $F_{sim}(\boldsymbol{x}; t) = k_{sim}(t) \tilde{F}_{sim}(\boldsymbol{x}; t)$.

The force $F(\boldsymbol{x}; t)$ is equivalent to the score component responsible for the atomic positions $s_r(\boldsymbol{x}, t)$, yet so far we have largely ignored the question of how to evolve the elemental composition $\boldsymbol{z}$. In the next section, we present a method to optimise $\boldsymbol{z}$ without having to define and train an alchemical force $s_z(\boldsymbol{x}; t)$.

**Handling element swaps.** We can obtain an alchemical force from the similarity energy by differentiating it with respect to the elemental embedding $\boldsymbol{z}$.

$$s_z(\boldsymbol{x}; t) \overset{?}{=} -\nabla_z E_{sim}(\boldsymbol{x}; t) \tag{15}$$

To represent atomic environments, we use a pretrained model that initially encodes the elements as one-hot categorical vectors. If we used $s_z(\boldsymbol{x}; t)$ directly, the discrete element encoding would become a continuous vector. While diffusion models can handle both discrete and continuous element embeddings - they are specifically trained to do so - a model trained only on discrete embeddings may fail to generate reasonable atomic environments when applied on structures with continuous element embeddings. To avoid this issue, we instead turn to an evolutionary algorithm.

We use a modified version of Particle Swarm Optimisation (PSO)[47] to optimise the element composition $\boldsymbol{z}$. PSO explores a wide solution space using a population of particles that share information about the current best solution. We introduce a mutation phase into PSO for element swapping. Each round begins by creating copies of the current best solution with the lowest $E_{sim}$. We then mutate a fraction of atoms in each particle by changing their element. Atoms with the lowest local similarity have the highest mutation probability. In addition, one particle remains unchanged to allow the possibility that the original $\boldsymbol{z}$ is already optimal. After mutation, the particles evolve independently according to Equation (14). This scheme enables element swaps without requiring an explicit $s_z$ and helps to find overall lower energy solutions by exploring a wider configuration space.

The modified PSO scheme has a downside – it struggles with single-valence elements. If we allow swaps to group 1 or 17 elements, the generation process almost always converges to isolated gas molecules. Since these elements have only one nearest neighbour, minimising $E_{sim}$ is trivial by switching to, for example, hydrogen and forming molecular $H_2$. To prevent this behaviour, we remove group 1 and 17 elements from the reference data and forbid swaps to these

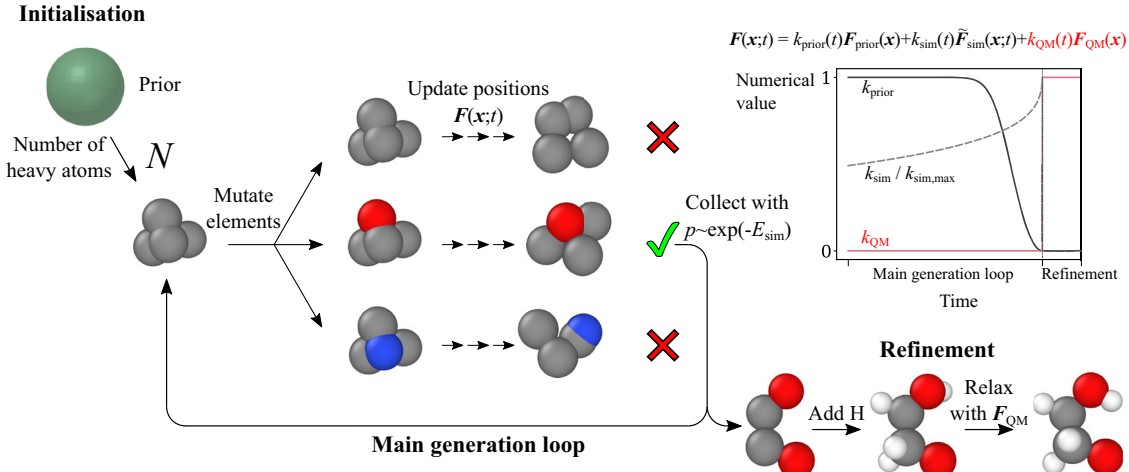

**Fig. 3 | Molecular generation using similarity kernels.** Generation begins by sampling initial atom positions from the selected prior (an isotropic Gaussian is used in the figure). The main loop then alternates between creating copies of the current configuration with randomly mutated elements and updating atom positions using a combination of two forces: a restorative force $\boldsymbol{F}_{\text{prior}}$ that maintains proximity to the original prior, and $\boldsymbol{F}_{\text{sim}}$ which maximises local similarity to the reference set (see main text for details). The upper right plot shows the relative strength of these two forces over time, gradually shifting from prior-dominated to similarity-dominated according to the schedule. After position updates, a single configuration is selected based on its similarity energy $E_{\text{sim}}$. In the refinement stage, hydrogens are added, and forces from a pretrained machine learning force field ($\boldsymbol{F}_{\text{QM}}$) are used to relax the structure. $k\cdot(t) =$ dimensionless prefactors that control the strength of the individual force components.

elements during PSO. We then add back hydrogen atoms separately after the PSO loop has finished.

Generation with explicit hydrogen atoms is an ongoing problem in the field. Including explicit hydrogen atoms generally reduces the performance of trained models[36,48] and increases cost without fundamentally improving the quality of the generated structures. As such, many groups choose to treat the addition of hydrogen as a post-processing task[49–51].

The PSO scheme we have developed is flexible and can be extended to optimise properties beyond element composition. If a fitness function can be defined for partially generated molecules with respect to a desired property, this score can be incorporated into the particle selection process alongside $E_{\text{sim}}$. Recent work by Cremer et al.[52] has explored a similar concept, framing it as importance sampling rather than PSO. Such extensions to our method could further enhance its utility in various molecular design tasks, from drug discovery to materials science.

**Full procedure.** Figure 3 illustrates the full generation procedure. The main generation loop uses the modified PSO and the force given by Equation (14) to generate a complete structure without hydrogen atoms. Since no hydrogen atoms are present, we set $k_{\text{QM}} = 0$ throughout this stage (the intermediate configurations without H appear extremely unsaturated, making physical forces less useful). After the main loop, we add back the hydrogen atoms and relax the geometry using physical forces only, $k_{\text{prior}} = k_{\text{sim}} = 0$.

The final structure of the generated molecule is primarily determined by the main generation loop. The refinement stage adds explicit hydrogen atoms and relaxes the molecule typically without further changes to the heavy atoms. Although we use purpose-trained models for refinement (see Similarity kernel details), the generative part of SiMGen does not require specialised models.

To evaluate our method, we used local environments extracted from increasing subsets of the QM9 dataset[53] to generate 1.5k molecules using each subset. Reference sets were created by first sampling up to two molecules uniformly based on the number of heavy atoms, then filling up to the required subset size via random sampling. We identify stable molecules by converting the generated point clouds

into SMILES format using Open Babel[54] and checking if they can be sanitised using RDKit[55].

Table 1 demonstrates that SiMGen can match the performance of trained models for unconditional generation of QM9-like molecules using just 256 reference molecules. As shown in the table, the effect of locality is particularly pronounced in the similarity-based novelty (Sim Novelty column) – by operating on a local environment level, SiMGen can generate molecules that are, on average, more distinct from the whole of QM9 than models trained on fully connected graphs. In addition, SiMGen generates a higher proportion of easily synthesizable molecules (Synthetic Accessibility Score < 3) compared to both trained models and the reference data. We hypothesise that this is due to the machine learning force field (ML-FF) used for refinement occasionally relaxing highly strained structures into more stable configurations.

These results - and experiments showing molecules generated with SiMGen match the QM9 energy distribution (SI Section S3) - show that SiMGen can effectively generate small molecules similar to a reference distribution. Next, we examine SiMGen's performance as we increase the size of the generated molecules.

**Scaling to larger molecules.** To analyse how SiMGen's performance depends on the size of the generated molecule, we generated a series of molecules ranging from 5 to 50 heavy atoms. Figure 4A illustrates the fraction of valid atoms versus the number of heavy atoms at the start of generation. As expected from a local generator, the percentage of valid atoms remains roughly constant at 99% before hydrogen addition, even when generating structures much larger than those in the reference set. However, after hydrogen addition, the fraction of atoms with valid valences slowly decreases as the size of the generated molecules increases.

One reason for the decrease in performance is the choice of the spatial prior. As the number of atoms increases, the shape estimated from QM9 molecules (a Gaussian with $\Sigma = \text{diag}(1., 1.4, 2.6)$, see SI Section S6 for details) forces atoms at the centre of the generation into high-density arrangements, often resulting in multiple quaternary carbon centres. Such configurations frequently cause errors during hydrogenation, leading to a low fraction of valid molecules. Figure 4B

**Table 1 | SiMGen matches the performance of trained diffusion models on unconditional generation using just 256 reference molecules**

| Metrics / % | Mol stable (↑) | Unique (↑) | | Novel (↑) | | SA distribution | | | Connected (↑) |
|---|---|---|---|---|---|---|---|---|---|
| | | SMI | Sim | SMI | Sim | SA < 3 | 3≤ SA < 6 | SA ≥6 | |
| QM9[53] | 100.0 | 99.9 | 14.9 | – | – | 9 | 87 | 4 | 100 |
| EDM[35] | 85.3 | 99.2 | 83.7 | 77.8 | 37.4 | 7 | 88 | 5 | 99.7 |
| MiDi[48] | 97.5 | 97.6 | 77.1 | 67.5 | 26.0 | 8 | 77 | 15 | 99.9 |
| SiMGen / 8 | 88.6±.2 | 98.6±.1 | 93.6 ±.2 | 100. | 75.8 ±.3 | 8 | 90 | 2 | 98.3±.0 |
| SiMGen / 64 | 92.6±.2 | 96.7±.2 | 87.8 ±.2 | 100. | 70.7±.2 | 10 | 87 | 3 | 98.6±.0 |
| SiMGen / 256 | 94.1±.2 | 97.5±.2 | 84.±.6 | 99.9 | 68.±.5 | 12 | 85 | 3 | 99.2±.0 |

Uniqueness and novelty are measured by 1) fraction of unique SMILES, and fraction of distinct SMILES from the QM9 dataset (SMI column), or 2) by Tanimoto similarities (Sim column, see SI Section S1 for details). Synthetic accessibility score (SA) of < 3 indicates generally easy-to-synthesise molecules; bioactive molecules and natural products tend to have a SAS of ≤ 6, and a higher score suggests a difficult synthesis. Numbers next to SiMGen (8, 64, 256) indicate the number of reference molecules used during generation. These scores were calculated using an RDKit[55] implementation of the algorithm described in ref. 87. Uncertainties were estimated by jackknife resampling. SMI = SMILES, Sim = Similarity.

reveals that using a more open prior (e.g., a flat disc or a long line) increases the fraction of valid molecules to ≈ 0.6 even for structures with 50 heavy atoms.

While SiMGen's local approach offers flexibility in incorporating spatial priors of arbitrary shapes, as we will demonstrate in the next section, it also inherently limits its ability to capture the global spatial characteristics of larger molecules. This is a fundamental constraint of purely local models – they lack the capacity to perceive long-range dependencies. Although the fraction of valid molecules generated by SiMGen could potentially be improved by developing more sophisticated methods for sampling valid molecular shapes, a more promising and practically relevant direction lies in exploring hybrid architectures. Such architectures, combining a global component to guide the overall molecular shape with a local component like SiMGen for detailed structure generation, could offer a more robust solution for generating complex and, importantly, synthesizable larger molecules. Therefore, we envision SiMGen not as a standalone solution, but as a powerful component potentially integrated into hybrid architectures or used alongside other methods addressing these broader chemical constraints, an approach we investigate further in Section II D by combining SiMGen with a trained diffusion model.

### Shape control

A key advantage of SiMGen lies in its ability to initialise generation from arbitrary prior distributions. This, combined with the local nature of the similarity kernel, grants precise control over the shape of generated structures, enabling the creation of molecules with complex geometries or specific spatial constraints. Using the octa-acid system as a case study, we show how this shape control can be leveraged when designing binders.

**Anisotropic Gaussian prior.** We start by extending the standard Gaussian to the anisotropic case with diagonal covariance $N(\boldsymbol{r}; \boldsymbol{0}, \boldsymbol{\Sigma})$. The probability density for a multivariate Gaussian is:

$$N(\boldsymbol{r}; \boldsymbol{0}, \boldsymbol{\Sigma}) \sim \exp\left(-\frac{1}{2}\boldsymbol{r}^{\mathrm{T}}\boldsymbol{\Sigma}^{-1}\boldsymbol{r}\right) \quad (16)$$

Taking the energy perspective of probability, we get $E_{\mathrm{prior}} = \boldsymbol{r}^{\mathrm{T}}\boldsymbol{\Sigma}^{-1}\boldsymbol{r}/2$ and the force $\boldsymbol{F}_{\mathrm{prior}} = -\nabla_{\boldsymbol{r}}E_{\mathrm{prior}} = -\boldsymbol{\Sigma}^{-1}\boldsymbol{r}$. To use this in generation, we initialise atom positions from $N(\boldsymbol{x}; \boldsymbol{0}, \boldsymbol{\Sigma})$ and apply $\boldsymbol{F}_{\mathrm{prior}}$ in equation (14).

Figure 5 shows that varying the covariance, $\boldsymbol{\Sigma}$, has a dramatic impact on the shape of the generated molecules. Elongated priors create molecules with long aliphatic chains, whereas flattened priors yield conjugated, planar structures.

**Point cloud prior.** We can construct a prior with an arbitrary shape via a point cloud. Consider a collection of $N$ Gaussians with means $\{\boldsymbol{\mu}_1, ..., \boldsymbol{\mu}_N\}$ and shared covariance $\boldsymbol{\Sigma}$. To sample this distribution, we first randomly select a point from the point cloud and then sample $N(\boldsymbol{\mu}_i, \boldsymbol{\Sigma})$. The density associated with this prior is given by Equation (17).

$$f(\boldsymbol{x}) = \sum_i^N \exp\left(-\frac{1}{2}(\boldsymbol{x} - \boldsymbol{\mu}_i)^{\mathrm{T}}\boldsymbol{\Sigma}^{-1}(\boldsymbol{x} - \boldsymbol{\mu}_i)\right) \quad (17)$$

The energy for this density is $E = -\log f(\boldsymbol{x})$. Fortunately, we can take the derivative of this energy analytically with the resulting force given by equation (18a).

$$\boldsymbol{F}_{\mathrm{prior}} = -\sum_i^N w_i \boldsymbol{\Sigma}^{-1}(\boldsymbol{x} - \boldsymbol{\mu}_i), \quad (18a)$$

$$w_i = \frac{\exp\left(-\frac{1}{2}(\boldsymbol{x} - \boldsymbol{\mu}_i)^{\mathrm{T}}\boldsymbol{\Sigma}^{-1}(\boldsymbol{x} - \boldsymbol{\mu}_i)\right)}{\sum_i \exp\left(-\frac{1}{2}(\boldsymbol{x} - \boldsymbol{\mu}_i)^{\mathrm{T}}\boldsymbol{\Sigma}^{-1}(\boldsymbol{x} - \boldsymbol{\mu}_i)\right)}. \quad (18b)$$

In practice, we use

$$w_i = \exp(-|\boldsymbol{x} - \boldsymbol{\mu}_i|)/\sum_i \exp(-|\boldsymbol{x} - \boldsymbol{\mu}_i|)$$

for computational simplicity, and have not observed any adverse effects on the generation procedure. Note that this derivation is analogous to equations for the similarity force.

In Fig. 6, we combine the point cloud prior with a flattened multivariate Gaussian to generate a sequence of macrocycles with increasing radii. The kernel's reference data contains environments from structures only up to nine heavy atoms, whereas the generated macrocycles range from 45 to 111 heavy atoms. Due to the locality of the kernel, we are able to generate structures much larger than those in the reference data. Although a local builder will eventually make a mistake when generating a very large molecule, these results showcase how a local model could be the backbone for generating large complex structures.

**Case study: Octa-acid binders.** Constrained generation is crucial for many applications in chemistry, such as designing new protein ligands or linking fragments. Our method can accommodate these constraints by incorporating stationary atoms or fragments into the simulation box. While maximising local similarity, these elements are naturally integrated into the final structure. However, due to the method's local nature, preplaced atoms can only influence generation if they are

**A**

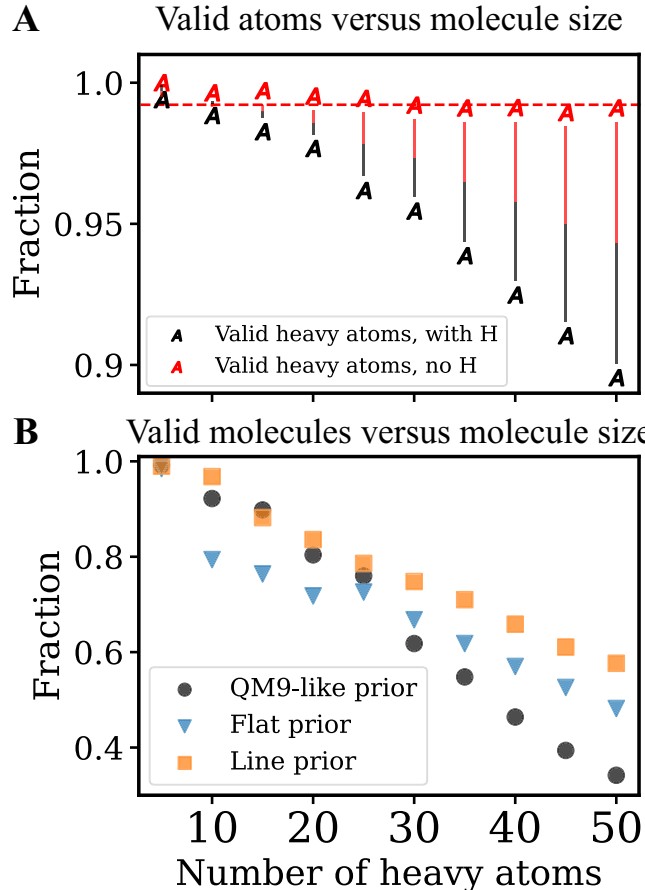

**Fig. 4 | The choice of spatial prior determines the validity of large generated molecules. A** Atom validity is independent of molecule size. Valid atoms are defined as atoms whose number of closest neighbours ≤ their natural valence. A molecule is valid if all its atoms are valid. Before the hydrogenation step, atom validity is 99%, after adding H atoms, the validity slowly decreases but remains above 90%. **B** Large molecules require different priors for improved validity. More open priors, i.e., lines or flat discs, improve the fraction of valid molecules when generating large structures. $n = 500$ for each point.

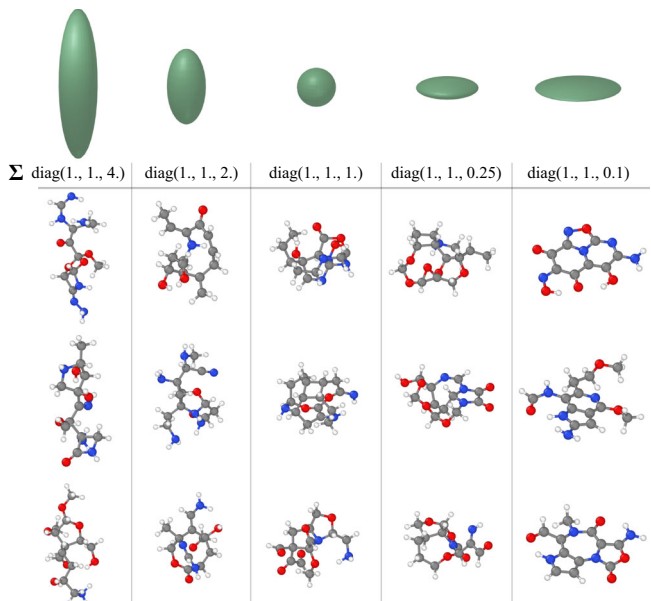

**Fig. 5 | Effect of the covariance matrix (Σ) on generated structures.** Ellipsoids at the top illustrate the priors arising from each covariance matrix. The choice of prior significantly affects the generated structure, producing geometries ranging from elongated to flattened.

within the receptive fields the atoms being optimised. Thus, constrained generation is most effective when combined with a point cloud prior that guides generation toward preplaced fragments.

As a proof-of-concept, we applied SiMGen to design binders for Octa-acid (OA), a supramolecular cage used in the SAMPL6 host-guest binding challenge[56]. OA's small size, conformational rigidity, and ability to bind small molecules make it a useful toy model for a protein binding pocket.

We constructed the prior using the eight guest molecules from the SAMPL challenge, superimposing their atoms to generate a point cloud (Fig. 7A). We then generated 500 molecules using either 256 QM9 molecules or the known OA binders as the reference set. Note that the OA cage was in the simulation box during the generation, meaning the similarity energy of the OA atoms closest to the binding pocket directly influenced the generation. After filtering out fragmented molecules, we redocked them using Glide[57].

Figure 7 B shows the cumulative distribution of docking scores for both approaches, alongside the scores for the original SAMPL binders. Molecules generated with the larger QM9-based reference set had slightly better docking scores, likely because the kernel requires more than eight configurations to reach best performance. Notably, when generating with the smaller reference set, SiMGen recovered one of the original SAMPL binders. Figure 7C displays the top two generated molecules.

Overall, this approach presents a simple and data efficient way to generate molecules conditioned on a binding pocket. The prior constructed from known binders (or other computational methods[58] in their absence) provides a strong positive bias for generating molecules that physically fit in the pocket – we did not observe any clashes with the OA cage during generation. Even though OA is a toy system, these results demonstrate the potential of spatially resolved similarity kernels for conditional generation.

**Interactive generation with ZnDraw.** Although generative tools are powerful, they can be difficult to set up and use for non-specialists. To facilitate the use of SiMGen and to engage the broader chemistry community, we have developed the ZnDraw software package[59]. ZnDraw provides functionality for visualising, modifying, and analysing atomistic systems. For generative modelling, users can load structures into ZnDraw, interactively specify 3D point cloud priors, and perform generation driven by these priors (Fig. S3 depicts how generation can be controlled via the ZnDraw interface). The flexibility to create custom prior shapes makes ZnDraw well-suited for users to design priors tailored to their specific applications. As an example, Fig. 8 demonstrates ZnDraw's interactive workflow to link two molecular fragments sourced from ref. 2. We provide an online version of ZnDraw with working generative modelling at https://zndraw.icp.uni-stuttgart.de.

**Combining SiMGen with Trained Models**
Having demonstrated SiMGen's effectiveness for generating small molecules with shape control and local conditioning, we now turn to addressing a key challenge for purely local models. While SiMGen's similarity energy captures local chemical patterns, many critical applications, particularly in drug discovery, necessitate satisfying global constraints. For instance, Lipinski's rule of five[60] includes a constraint on the total number of hydrogen bond donors in a molecule - a global property.

To expand SiMGen's practical utility and overcome this limitation, we demonstrate its effective integration with an existing diffusion model (MolDiff[61]), enabling fragment-biased generation. Fragment-

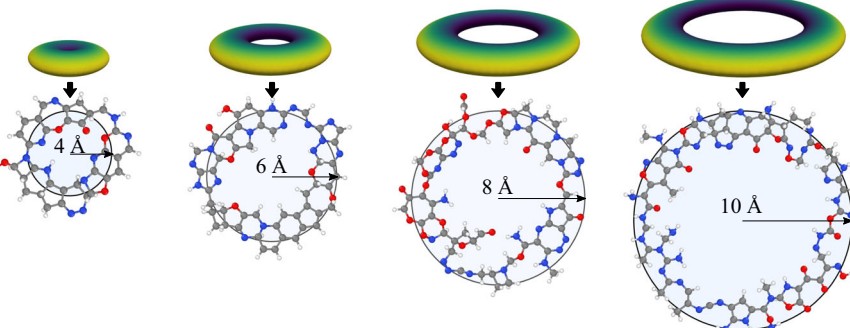

**Fig. 6 | Point cloud prior enables generation in an arbitrary shape.** Macrocycle-like structures can be generated by combining a circular prior with a flat covariance. Specifically, the point cloud prior is constructed by placing 100 equidistant points on a circle of radius $r = \{4, 6, 8, 10\}$ Å., each point of the prior has a covariance matrix diag(1., 1., 0.5). Colour in the prior corresponds to the distance from the centre.

constrained generation[62,63] aims to create molecules that incorporate pre-defined structural fragments. This is particularly relevant in drug discovery, where combining multiple small molecules, i.e., fragments, into a single strong binder is a well-established approach[64,65]. Our proposed approach offers a convenient and data-efficient route to guide any atomistic score-based generative model towards desired fragments.

**Guiding a trained model.** Score-based models can be guided by incorporating additional information into their learned score function, $s(\boldsymbol{x}, t)$[66]:

$$s_{\text{gui}}(\boldsymbol{x}, t) = s(\boldsymbol{x}, t) + k_{\text{gui}}(t)\nabla_{\boldsymbol{x}} \log p(c|\boldsymbol{x}).$$

Here, $c$ represents the desired property, and $p(c|\boldsymbol{x})$ is the probability of a configuration $\boldsymbol{x}$ possessing that property. In energy-based guidance[67], the probability is modelled via an energy function $p(c|\boldsymbol{x}) \propto \exp(-E(\boldsymbol{x}))$.

If the desired property is a similarity to some reference set, then SiMGen offers a directly compatible, differentiable energy function, $E_{\text{sim}}(\boldsymbol{x}, t)$. This leads to a guided score:

$$s_{\text{gui}}(\boldsymbol{x}, t) = s(\boldsymbol{x}, t) - k_{\text{gui}}(t)\nabla_{\boldsymbol{x}}E_{\text{sim}}(\boldsymbol{x}, t) \tag{19}$$

$$= s(\boldsymbol{x}, t) + k_{\text{gui}}(t)\boldsymbol{F}_{\text{sim}}(\boldsymbol{x}, t) \tag{20}$$

Thus, any atomistic score-based model can be guided by incorporating the similarity force, $\boldsymbol{F}_{\text{sim}}(\boldsymbol{x}, t)$, into its score function as defined in Eq. (20).

Alternatively, energy-based guidance can be implemented without directly modifying the score. During batch generation, where $n = m \cdot k$ molecules are generated, we can divide them into $k$ groups of $m$ molecules each and apply minibatch importance sampling. Every $N_{\text{IS}}$ steps, each minibatch $\mathcal{B} = \{\boldsymbol{x}_1, \ldots, \boldsymbol{x}_m\}$ is resampled with replacement according to:

$$\mathcal{B} \leftarrow \left\{\boldsymbol{x}_i | \boldsymbol{x}_i \sim \exp\left(-\frac{E_i}{T}\right), i = 1, \ldots, m\right\} \tag{21}$$

Here, $T$ is a temperature parameter controlling the acceptance of higher energy structures. This approach is closely related to our PSO scheme and to existing importance sampling schemes that aim to optimise other chemical properties[52].

Importance sampling allows the use of non-differentiable energy functions and avoids the need for gradient calculations. However, it can reduce diversity within minibatches and, as we will show, may be insufficient when guiding towards rare properties.

**A** Construction of prior from known binders

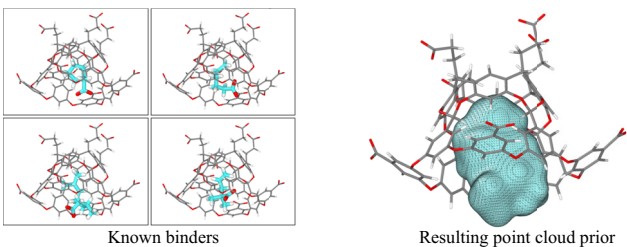

Known binders    Resulting point cloud prior

**B** Glide score distribution of SiMGen generated molecules

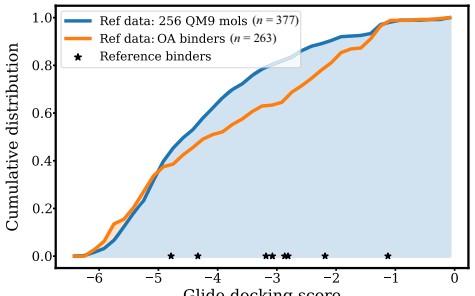

**C** Molecules generated with SiMGen and redocked with Glide.

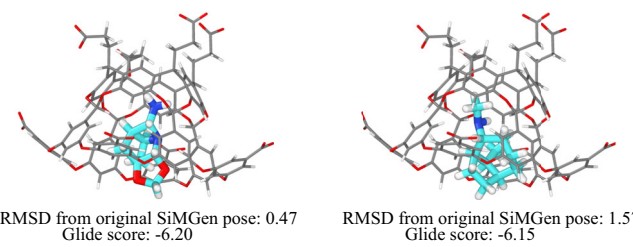

RMSD from original SiMGen pose: 0.47  RMSD from original SiMGen pose: 1.57
Glide score: -6.20       Glide score: -6.15

**Fig. 7 | Conditional generation enables binder design. A** Atoms from known binders are superimposed to create a point cloud prior. **B** Black stars indicate the docking scores of the SAMPL6[56] guests used to construct the prior. Note that some generated molecules did not dock into the OA cage; the actual number of docked molecules is indicated in the legend. **C** Examples of binders generated with SiMGen. The OA cage was explicitly present during generation. OA = Octa Acid, RMSD = Root Mean Square Deviation.

Guidance of generative models is an active research area; for a more comprehensive discussion, see ref. 68.

**Inverse Sum Order.** The original similarity energy formulation (Eq. (11)) encourages configurations where each atom closely matches at least one reference environment. This arises from the structure of the equation: the logarithm requires the inner sum to be non-zero for each

atom $i$, as otherwise $\lim_{x \to 0} \log(x) = -\infty$. Consequently, to minimise the energy, each atomic environment in the generated configuration tends to align with some environment in the reference set.

While effective for unconditional generation, this formulation is not ideal for guiding generation towards a specific chemical space. Specifically, the original energy function does not prevent multiple atoms in the generated configuration from being driven towards matching the same reference environment.

A simple modification to address this is to swap the order of summation, as shown below:

$$E_{\mathrm{sim}}(\boldsymbol{x}; t) = -\sum_{i \in \boldsymbol{x}} \log\Bigg( \overset{\text{swap}}{\sum_{j \in \mathcal{D}_{\mathrm{ref}}} k(\boldsymbol{\chi}_i, \boldsymbol{\chi}_j; t)} \Bigg), \quad (22)$$

$$E_{\mathrm{inv}}(\boldsymbol{x}; t) = -\sum_{j \in \mathcal{D}_{\mathrm{ref}}} \log\Bigg( \sum_{i \in \boldsymbol{x}} k(\boldsymbol{\chi}_i, \boldsymbol{\chi}_j; t) \Bigg). \quad (23)$$

This small change fundamentally alters the behaviour of the similarity energy. With the inverse sum order, the energy $E_{\mathrm{inv}}(\boldsymbol{x}; t)$ is minimised when all reference environments find a match within the generated configuration. This makes the inverse formulation well-suited for fragment-biased generation, where the objective is to ensure that the generated molecules contain specific, predefined fragments represented in the reference set.

**Case study: Penicillin.** To showcase SiMGen's guidance capabilities, we combined it with MolDiff[61], a diffusion model pretrained on the GEOM-Drug dataset[69]. Our objective was to bias MolDiff towards generating molecules containing the penicillin core, which features a $\beta$-lactam ring fused with a thiolane ring - iconic motifs of penicillin antibiotics. As such, we used this core fragment as the reference set for SiMGen.

Figure 9 A compares molecules generated by unguided MolDiff versus MolDiff guided by SiMGen. The highest similarity molecules from the guided generation consistently feature both the $\beta$-lactam and thiolane/thiophene ring fragments of the penicillin core. In contrast, among 5000 molecules generated by the unguided MolDiff, none contained both of these fragments.

Figure 9 B illustrates the impact of different guidance approaches on the distribution of generated molecules. While minibatch importance sampling alone had a negligible effect - likely due to the scarcity of the penicillin core in MolDiff's training data (only 0.1% of training molecules contained a $\beta$-lactam fragment) - score-based guidance using the inverse sum similarity energy proved significantly more effective. The score-based guidance resulted in an 8-fold increase in the generation of molecules containing $\beta$-lactams and a 3-fold increase in those with thiolane ring derivatives (see supplementary Section S4 for details). The limited effectiveness of importance sampling underscores that a reasonable representation of the target property within the generative model's prior distribution is crucial for importance sampling to be an effective guidance mechanism.

While effective, SiMGen guidance introduces a degree of fuzziness. This fuzziness stems from SiMGen's reliance on smooth, local atomic environment descriptors. For instance, the similarity kernel might consider modifications to ring sizes or the splitting of fused rings as minor perturbations, while medicinal chemists would recognise these as significant structural changes. This inherent flexibility can be partially mitigated by more aggressive importance sampling settings (larger minibatches and lower temperature), but at the expense of sample diversity. The optimal balance of generation parameters will ultimately depend on the specific application.

Finally, as with any guidance method, the relative strength of the guidance force compared to the original score function requires

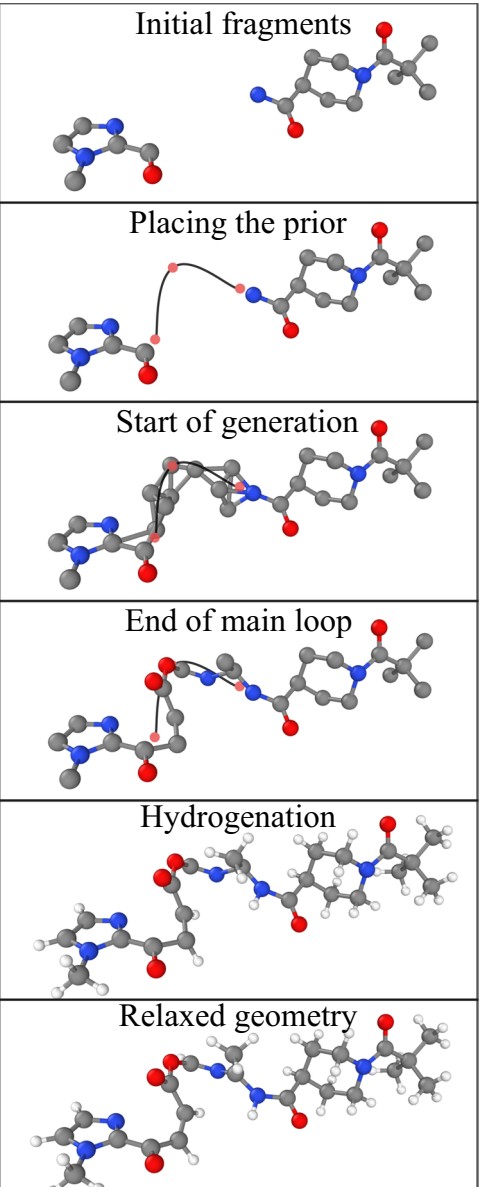

**Fig. 8 | Interactive generation with ZnDraw.** From top to bottom: the different stages of generation using the interactive tool. The black silhouette shows original atom positions.

careful tuning. Stronger guidance enhances its effectiveness but also increases the risk of pushing the generation process outside the original training distribution, leading to generation failure. Furthermore, as highlighted by the importance sampling example, the prior distribution remains a significant factor, limiting the extent to which guidance can reshape the generated output.

## Methods
### Similarity kernel details
The following sections give more information on the main components of the generation and clarify the use of trained models. Briefly, we use a pretrained MACE model to generate representations for the kernel and, during refinement, to swap elements and relax the geometry. In addition, we use a trained model for the hydrogenation step; however, this is a design choice, not a requirement. The hydrogenation model could be replaced with any standard method, e.g., using bond lengths to infer missing hydrogen atoms.

**A**

MolDiff

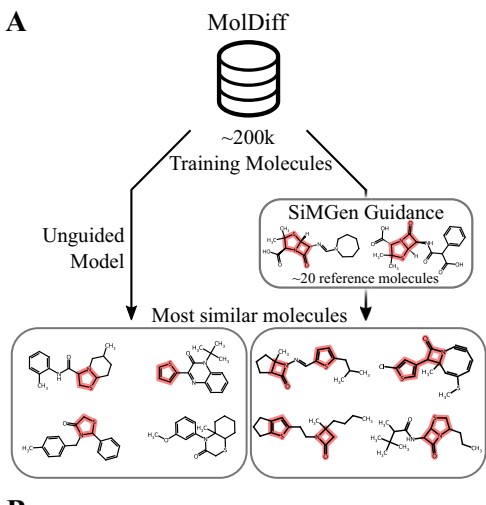

~200k
Training Molecules

Unguided Model

SiMGen Guidance

~20 reference molecules

Most similar molecules

**B**

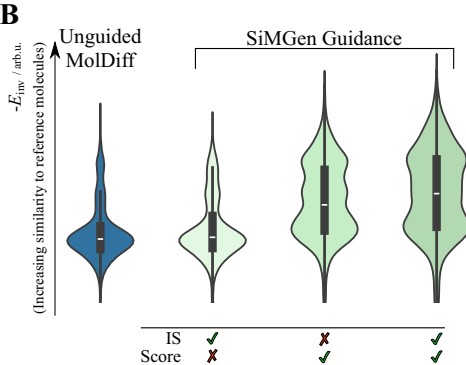

Unguided MolDiff

SiMGen Guidance

$-E_{inv}$ / arb.u.

(Increasing similarity to reference molecules)

| | | | |
|---|---|---|---|
| IS | ✓ | ✗ | ✓ |
| Score | ✗ | ✓ | ✓ |

**Fig. 9 | SiMGen enables fragment-biased generation. A** Combining MolDiff[61], a trained diffusion model, with SiMGen and a small dataset of reference fragments biases the generation towards molecules carrying these fragments. The depicted molecules are the most similar structures to the penicillin core, as measured by the lowest $E_{inv}$ (a SiMGen similarity energy with inverse summation order; see main text for details). **B** Distributions of similarity to penicillin core using importance sampling (IS) and/or score-based guidance as indicated by the table. Each distribution is based on $n \approx 200$ molecules. In the boxplots, the white line indicates the median, the thick box outlines represent the lower and upper quartiles (25th and 75th percentiles), and the whiskers extend to the most extreme data points within 1.5 times the interquartile range.

**MACE features.** We extract the features from a MACE[31,70] model trained on the SPICE dataset[71] containing 1 million molecules of 3 to 100 atoms. The MACE model is a many-body equivariant message passing neural network (MPNN). We summarise the key steps of the construction of the MACE descriptors.

The first step in MACE is to construct the local neighbourhood of an atom based on a cutoff distance, $\mathcal{N}(i) = \{j | r_{ij} \leq r_{cut}\}$. Then two body information is encoded in a one particle basis as a product of radial basis function $R$, spherical harmonics $Y_l^m$ and node features $h_j$, displayed in Eq. (24) where $\hat{r}_{ij}$ denotes the relative positions. The node features are initialised as one hot chemical element, $h_j^{(0)} = z_j$. The one particle basis is summed over the neighbourhood to achieve permutation invariance in Eq. (25), after which $(v+1)$-body features are formed in (26) by taking the tensor product of the atomic basis $A_{i, kl_3 m_3}^{(s)}$ with itself $v$ times. The tensor product is symmetrised by contracting the Clebsch Gordan coefficients $\mathcal{C}_{\eta, lm}^{LM}$ where $\eta$ enumerates all possible symmetric couplings to form the symmetrized basis $B_{i, \eta kLM}^{(s)}$. A message is formed as a learnable combination of the symmetrised basis in Eq. (27). The next node features are for the step $(s+1)$ formed by applying a linear update function on the message in (28). This operation is

repeated $S$ time, always reusing previously constructed node features.

$$\phi_{ij, k\eta_1 l_3 m_3}^{(s)} = \sum_{l_1 l_2 m_1 m_2} C_{\eta_1, l_1 m_1 l_2 m_2}^{l_3 m_3} R_{k\eta_1 l_1 l_2 l_3}^{(s)}(r_{ij}) \times \\ \times Y_{l_1}^{m_1}(\hat{r}_{ij}) \bar{h}_{j, kl_2 m_2}^{(s)} \tag{24}$$

$$A_{i, kl_3 m_3}^{(s)} = \sum_{\bar{k}, \eta_1} w_{k\bar{k}\eta_1} \sum_{j \in \mathcal{N}(i)} \phi_{ij, \bar{k}\eta_1 l_3 m_3}^{(s)} \tag{25}$$

$$B_{i, \eta kLM}^{(s)} = \sum_{lm} \mathcal{C}_{\eta, lm}^{LM} \prod_{\xi=1}^{v} A_{i, kl_\xi m_\xi}^{(s)} \tag{26}$$

$$m_{i, kLM}^{(s)} = \sum_v \sum_{\eta_v} W_{z_i \eta_v, kL}^{(s), v} B_{i, \eta_v kLM}^{(s), v} \tag{27}$$

$$h_{i, kLM}^{(s+1)} = \sum_{\bar{k}} W_{kL, \bar{k}}^{(s)} m_{i, \bar{k}LM}^{(s)} + \sum_{\bar{k}} W_{kz_i L, k}^{(s)} h_{i, \bar{k}LM}^{(s)} \tag{28}$$

Experiments with the trained diffusion model (Section II A) show that being able to handle overlapping atoms is important to the generation process. In general, MACE features vary smoothly with changes in the atomic coordinates; however, this is less true when the interatomic distances approach zero. The reason for this is twofold: one, there is little training data in this region, and two, the features must encode the rapidly increasing repulsive forces as atoms start to overlap. We can avoid this by introducing a distance transform in equation (24).

$$\phi_{ij, k\eta_1 l_3 m_3}^{(s)} = \sum_{l_1 l_2 m_1 m_2} C_{\eta_1, l_1 m_1 l_2 m_2}^{l_3 m_3} R_{k\eta_1 l_1 l_2 l_3}^{(s)}(\tilde{r}_{ij}) \times \\ \times Y_{l_1}^{m_1}(\hat{r}_{ij}) \bar{h}_{j, kl_2 m_2}^{(s)} \tag{19a}$$

$$\tilde{r}_{ij} = r_{min} + \max(0, r_{ij} - r_{min}) \tag{19b}$$

Here we used $r_{min} = 0.75$ Å. This distance transform ensures the MACE features remain well-behaved even when atoms are fully overlapping, which occurs frequently at the start of the generation.

The pretrained MACE model used here has two layers, 96 channels (number of $k$), a maximum angular resolution of $l_{max} = 3$ (maximum $l_3$ in the equation (25)), and message equivariance of $L = 0$ and correlation of $v = 3$ at each layer. For the kernel, we extract the invariant scalar node features of the first layer $h_{i, k00}^{(0)}$. They are many-body descriptors of the local environments of the atom $i$ and contain information within a 5.0 Å around each atom.

The MACE architecture ensures that the extracted features follow the necessary symmetries. Moreover, the obtained representations are differentiable, which is a requirement when using the kernel as a generative model.

**Generation details.** The main generation loop combines the forces in Equation (14), a modified Heun sampler, and PSO to generate structures.

**The force** schedule used during generation is given by Equation 29. As stated in the main text, the QM force is only used for refinement.

$$k_{prior}(t) = \tanh(20t^2) \tag{29a}$$

$$k_{sim}(t) = \frac{1}{\sigma(t)^2} = 119(1 - (t/10)^{1/4}) + 1 \tag{29b}$$

Although these specific schedules worked well, in general, any functions that provide the right balance between the two forces and significantly decrease the kernel width by the end of the generation could be effective.

In addition to the schedule, $\boldsymbol{F}_{\text{prior}}$ requires a multiplier dependent on the size of the generated structure. $|\boldsymbol{F}_{\text{prior}}|$ effectively creates a volume in which atoms can move freely. Without atom number dependence, the restorative force is either too strong, squeezing many atoms into too small a volume, or too weak, allowing atoms to dissociate. Therefore, $\boldsymbol{F}_{\text{prior}}$ is multiplied by a factor $\propto 1/n$ during generation, ensuring that the magnitude of the restorative force matches the size of the molecule.

Although $E_{\text{sim}}$ is generally well behaved, its repulsive regions can contain small minima, leading to atom overlap during generation. To avoid this, we add a soft short-ranged repulsive term to $E_{\text{sim}}$:

$$E_{\text{repulsive}} = \frac{1}{2}\sum_{ij} \exp(-\alpha r_{ij}) \tag{30}$$

$\alpha$ is a hyperparameter controlling the range of the repulsion.

**The sampler** used with the similarity kernel builds off Algorithm 2 from ref. 46. We make two modifications. First, we set a minimum noise level added to the positions at each sampler step. Second, we use a custom time step schedule that is linear for the first $N/2$ steps, then decays exponentially for the remaining steps.

**In PSO,** $N_{\text{particles}}$ copies of the current configuration are created. Of these, $N_{\text{particles}}-1$ are mutated by swapping elements, while one remains unchanged. Each mutation changes the element of $\lceil 0.2 \cdot n_{\text{atoms}}\rceil$ atoms, chosen based on their local similarity to the reference set:

$$p_{\text{mutate}}(i) \propto \exp(\beta \log f(\chi i; t)) \tag{31}$$

Here $\beta \propto \exp(-t)$ is an inverse temperature. This temperature annealing encourages exploration of diverse compositions in the initial stages of generation and focuses the swapping on the lowest similarity environments at the end of generation. For this work, the element swaps are limited to C, N, and O. After mutation, the particles evolve independently for $n_{\text{freq}}$ sampler steps. The next round of PSO begins by selecting one particle with $p \propto \exp(-\beta E_{\text{sim}})$, keeping it as the starting point for the next mutations. For the examples here, we used $N_{\text{particles}} = 10$ and $n_{\text{freq}} = 2$.

**Refinement details.** The refinement stage has three steps:
1. Addition of hydrogen atoms
2. Relaxation
3. Valence check

**Hydrogenation** Typically, when generating without hydrogen, generative models use bond lengths to infer bond orders and, in turn, the number of hydrogens required to satisfy the valence of each atom[35,49–51]. This approach is also applicable here: We can use the bond lengths to determine how many hydrogens to add and create valid molecules. However, this method fails for molecules with extensive conjugation, such as benzene, where the bond order falls between discrete values.

Instead, we trained a MACE model to predict the number of additional bonds required to satisfy the valence (https://github.com/RokasEl/hydromace). The predicted number of hydrogen atoms is added to the structure using rejection sampling for the initial positions. Then, the pretrained MACE model is used to correct the positions of the hydrogen atoms.

**Relaxation** After hydrogen relaxation, we employ the pretrained MACE force field with the LBFGS algorithm to relax the entire structure.

**Valence check** To account for potential bonding changes during relaxation, we use the hydrogenation model to check if additional hydrogen atoms are needed. If so, they are added and the hydrogen atoms are re-relaxed.

**Relation to score-based generative modelling.** Following the gradient of a similarity kernel is an expedient approach to generating new environments that resemble the data used to parameterise the kernel. However, its relationship to the diffusion modelling paradigm we originally set out to emulate is not obvious.

In fact, the score and the kernel density estimate are closely related. Vincent[11] have shown that the denoising score-matching objective is equivalent in expectation to learning the gradient of the kernel density estimate over the training data.

A key difference between SiMGen and vanilla score-based generative models is the space in which they operate. Typically, the score is learnt in the same space as the input data. For molecular generation, this corresponds to the positions and the element types of the input atoms. In contrast, SiMGen constructs a score based on similarity in a latent space.

More precisely, we construct the similarity kernel using the learnt local atomic representations from a pretrained MACE model, and it is the score of the time-varying probability distribution of latent representations that we obtain through the similarity gradient. By applying a generative process in a learnt representation space, our approach resembles that of latent diffusion[72,73]. However, SiMGen differs in that we follow the score in the latent space but apply updates at every step in the data space, that is, to the atomic coordinates. In standard latent diffusion, the representation is mapped to the data space only at the end of the diffusion process. By recomputing the latent representation from the data at every update step using the pretrained model, we ensure that we remain on a relevant manifold in the representation space.

### Energy-based diffusion model

The energy-based model uses a modified MACE architecture by adding a time encoding. Specifically, time is positionally encoded[74] and combined with the node features:

$$\tilde{h}_{i,k}^{(0)} = \text{MLP}(h_{i,k}^{(0)}, f(t)) \tag{32}$$

where $h_{i,k}^{(0)}$ are the original MACE features. $\tilde{h}_{i,k}^{(0)}$ replaces $h_{i,k}^{(0)}$ in subsequent uses within MACE.

To turn the modified MACE into a generative model, we use the preconditioning proposed by Karras et al.[46]. In this design, the noise schedule is linear $\sigma(t) = t$ and noised samples are created by adding noise without scaling.

$$\boldsymbol{x}_t = \{\boldsymbol{r}_t, \boldsymbol{z}_t\} = \{\boldsymbol{r}_0 + N(\boldsymbol{0}, \boldsymbol{I}t^2), \boldsymbol{z}_0 + N(\boldsymbol{0}, \boldsymbol{I}t^2)\} \tag{33}$$

Time $t$ is sampled from a log-normal distribution $\log t \sim N(P_{\text{mean}}, P_{\text{std}}^2)$. Denoised predictions are obtained by wrapping the score in a preconditioning scheme.

$$D_{\boldsymbol{r}}(\boldsymbol{x}_t, t) = c_{\text{skip}}(t)\boldsymbol{r}_t + c_{\text{out}}(t)\nabla_{\boldsymbol{r}}E(c_{\text{in}}(t)\boldsymbol{x}_t; c_{\text{noise}}(t)\cdot t) \tag{34a}$$

$$D_{\boldsymbol{z}}(\boldsymbol{x}_t, t) = c_{\text{skip}}(t)\boldsymbol{z}_t + c_{\text{out}}(t)\nabla_{\boldsymbol{z}}E(c_{\text{in}}(t)\boldsymbol{x}_t; c_{\text{noise}}(t)\cdot t) \tag{34b}$$

The loss is the sum of position and element denoising losses:

$$L = \frac{1}{\lambda(t)}\left(||\boldsymbol{r}_0 - D_{\boldsymbol{r}}(\boldsymbol{x}_t, \sigma(t))||^2 + ||\boldsymbol{z}_0 - D_{\boldsymbol{z}}(\boldsymbol{x}_t, \sigma(t))||^2\right) \tag{35}$$

The exact scaling $c_.(t)$ and weighting $\lambda(t)$ functions can be found in Table 1 in ref. 46. Molecules are generated by initialising from

$N(\mathbf{0}, \mathbf{I}\sigma(t_{max})^2)$ and using the stochastic sampler defined in Algorithm 2 in ref. 46.

The model used to generate the energy landscape shown in section II A was trained on 80% of the QM9 dataset for 300 epochs using the one-cycle learning rate policy[75]. The model used a cutoff of 10 Å (this corresponds to a global model), 16 radial basis functions, two interaction layers, 64 channels and message equivariance of $L = 1$ and correlation of $v = 3$ at each layer.

For computing the $E_{linear}$ and $E_{QM}$ baselines, we used PM6[76] with MOPAC[77] as a surrogate for the real QM energy.

## ZnDraw details

To enable interactive molecular generation, a visualisation of the system, together with the ability to draw 3D point cloud priors, is needed. Furthermore, a connection between the interface and HPC resources has to be established. We achieve this in ZnDraw by developing it as a web application that communicates through websockets using the socket.io standard. The server is built in Python and supports file input through the Atomic Simulation Environment[78] but can also read files that comply with H5MD[79]. The visualisation is realised through the JavaScript package three.js.

A hosted ZnDraw instance supports multi-user access, including sharing sessions for a real-time collaborative experience. Each session can be connected to one or more Python kernels for manipulating the data.

```
from zndraw import ZnDraw
vis = ZnDraw(url="<url>", token="<token>")
```

The connection to the SiMGen software is realised through ZnDraw's plugin interface. Methods to modify the scene with a given set of parameters can be added using Pydantic, defining the parameters that will be displayed through the user interface.

```
from pydantic import BaseModel

class MyModifier(BaseModel):
    parameter: float = 3.14

    def run(self, vis: ZnDraw): ...

vis.register_modifier(MyModifier)
```

ZnDraw can be installed locally through `pip install zndraw` on all standard operating systems and uses a command-line interface `zndraw <file>` to visualise molecular structures or interface with a local version of SiMGen.

Furthermore, it is possible to view remote data made available using the ZnTrack[80] package, as exemplified in the SiMGen live demo by the reference dataset and the hydrogenation model.

## Visualisation

Matplotlib[81] and Ovito[82] were used extensively to generate the figures in this work.

## Reporting summary

Further information on research design is available in the Nature Portfolio Reporting Summary linked to this article.

## Data availability

The pretrained MACE models used to generate the representations are available for free for academic purposes at https://github.com/ACEsuit/mace-off. The hydrogenation model and the precise sets of reference molecules are available at https://github.com/RokasEl/MACE-Models[83]. Finally, structures generated using SiMGen including baselines and case studies, and source data have been deposited in the Figshare repository and are available at https://doi.org/10.6084/m9.figshare.25447309[84].

## Code availability

• The code for SiMGen is available at https://github.com/RokasEl/simgen[85] under an MIT licence. • The code for SiMGen guidance of MolDiff is available at https://github.com/RokasEl/MolDiff-SiMGen[86] under an MIT licence. • The code for ZnDraw is available at https://github.com/zincware/ZnDraw[59] under an EPL-2.0 licence. The package can be installed via `pip install zndraw`. • The ZnDraw and SiMGen demo is hosted at https://zndraw.icp.uni-stuttgart.de.

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

## Acknowledgements

We thank Tamás K. Stenczel for initially proposing the idea of using similarity kernels for molecular generation, J Harry Moore for carrying out the docking of the Octa-acid binders, and Lars L. Schaaf for helpful discussions. R.E. and I.B. acknowledge support by the University of Cambridge Harding Distinguished Postgraduate Scholars Programme. S.W.N. acknowledges support from the European Union's Horizon 2020 research and innovation programme under the Marie Skłodowska-Curie Actions (Grant Agreement 945357) as part of the DESTINY PhD programme, as well as support from the European Union's Horizon 2020 research and innovation programme under Grant Agreement 957189 (BIG-MAP). C.H. and F.Z. acknowledge support by the Deutsche Forschungsgemeinschaft (DFG, German Research Foundation) in the framework of the priority programme SPP 2363, "Utilisation and Development of Machine Learning for Molecular Applications - Molecular Machine Learning" Project No. 497249646. C.H. and F.Z. acknowledge further funding through the DFG under Germany's Excellence Strategy - EXC 2075 - 390740016 and the Stuttgart Centre for Simulation Science (SimTech). Access to CSD3 was obtained through a University of Cambridge EPSRC Core Equipment Award EP/X034712/1. We acknowledge funding from UKRI under the UK Car-Parrinello HEC Consortium grant, with number EP/X035891/1.

## Author contributions

R.E. implemented SiMGen, trained the baseline and hydrogenation models, ran the analyses, and wrote the initial version of the manuscript. F.Z. implemented ZnDraw and wrote the section detailing ZnDraw. F.Z. and R.E. integrated SiMGen with ZnDraw and optimised the package to support multiple concurrent users. IB wrote the section about the MACE features. I.B. and D.P.K. provided guidance on using the pretrained MACE models. S.W.N. wrote the section connecting the similarity energy and force to score-based generative modelling. C.H. and G.C. supervised the project. All authors discussed the results, helped plan the project, and edited the final manuscript.

## Competing interests

G.C. has equity interest in Symmetric Group LLP that licences force fields commercially and also in Ångström AI. S.W.N. has financial interest and equity stake in Mirror Physics, a company working on AI and atomistic modelling. There are no more competing interests.
