## [Transparent Peer Review file · Nature Communications]

Zero Shot Molecular Generation via Similarity Kernels

Corresponding Author: Mr Rokas Elijosius

Version 0:

Reviewer comments:

Reviewer #1

(Remarks to the Author)

The paper "Zero Shot Molecular Generation via Similarity Kernels" introduces an innovative approach to molecular generation using a model named SiMGen. This model utilizes a time-dependent similarity kernel combined with descriptors from a pre-trained machine learning force field to facilitate molecule generation without additional training. The authors have also developed an interactive web tool to democratize access to their method, allowing broader engagement and practical application.

Strengths of the Paper:

1. The paper presents a novel approach to molecular generation by integrating similarity kernels with a zero-shot learning framework. This methodology is significant as it addresses the challenges of conditional molecule generation without retraining or fine-tuning the model on new datasets. However, the quality heavily depends on the quality of the pre-trained force field model.
2. Using point cloud priors to control the molecular shape introduces a novel way to incorporate user-specified constraints into the generation process. This aspect enhances the model's applicability to custom molecule design, where specific spatial configurations are critical, such as in drug design, where molecules must fit precisely within a target binding site.
3. The interactive web tool facilitates the practical use of the model and makes it accessible to a broader audience, including those without deep technical expertise in machine learning or programming.

Weaknesses and Limitations:

In my opinion, the main weaknesses of the paper are the lack of comprehensive validation and insufficient benchmarking. The paper does not provide empirical evidence of the model's performance. Below are a few suggestions on how to improve it:

1. Conduct a thorough quantitative analysis using metrics such as validity (how many generated molecules are chemically viable), uniqueness (diversity of the output), and novelty (comparison against known datasets) for unconstrained generation. These metrics are standard in assessing molecular generation models and will provide a solid foundation for evaluating the model's performance. If possible, compare SiMGen's performance with other state-of-the-art methods, including diffusion generative models, to clearly show where it stands in relation to existing technologies.
2. Demonstrate the model's capabilities for targeted molecule design in one or two case studies. In my opinion, conditional generation is the most important practical application of models for molecule generation. Case studies could include designing a drug molecule with desired biological activity or desired pharmacophore or materials with particular physical properties. This will greatly enhance the paper's relevance and applicability, showcasing the practical utility of the model beyond theoretical capabilities.

Although the proposed approach is novel, technically sound, and interesting, I think the paper cannot be accepted for publication in its current form as it lacks the necessary evaluation component to demonstrate the performance and capabilities of the proposed approach.

Minor remark:

Line 283: missing colon after "Low transferability"

(Remarks on code availability)

The code repository is well-organized, contains a README file, includes user instructions. The method is also installable as a Python package through pip, although I have not tried running the code or installing the package myself. The interactive web tool was also accessible and functioning as of April 15, 2024.

Reviewer #2

(Remarks to the Author)

Summary

The authors propose a score-based generation process using a time-varying local similarity kernel defined by a prior dataset and features extracted from a purpose pre-trained model, MACE. The idea proposed to solve the existing problem is good and seems reasonable in theory. However, the experimental design part needs some validation.

Strength

1) The proposed process does not require training, so it is not constrained by the prior distribution.

2) The ability of the proposed process to control the shape of the molecules generated according to an arbitrary prior was very impressive.

Weakness

1) One of the cores of the proposed algorithm is similarity kernels, which are defined by the RBF kernel. Similarity kernels require reference data, and in this paper, 259 QM9 molecules were used as reference data. The molecules generated are affected by how the reference data is organized; therefore, the authors should analyze the generation model's results according to the reference data configuration for the readers' understanding, e.g., how the molecule generation changes according to the reference data configuration, what is the sufficient number of reference data sizes needed for molecule generation, etc.

2) Also, the proposed score includes a term related to the prior. Therefore, it seems important to add an analysis related to the effect of the prior and similarity energy on the molecules generated. In this paper, there are only results where the molecules generated according to the prior can be controlled. Such an analysis would not only be of interest to the reader but would also make it easier to understand the proposed method.

3) The proposed algorithm is only compared to a baseline based on linear interpolation. More comparative experiments with various state-of-the-art methodologies are needed.

4) In the field of molecular generation (e.g., drug generation), one of the key concerns is the novelty of molecules generated by artificial intelligence or machine learning. The proposed algorithm shows very high atom environment similarity to the data used as reference. While this is not a direct measure of the similarity between molecules, it is worth checking that the molecules generated by the proposed algorithm are not only similar to known molecules.

5) The molecules generated by the proposed algorithm show a significant drop in the percentage of valid molecules from around 10 heavy atoms. It seems to be around 0.6-7 when hydrogen is included, but it would be invaluable to see more experimental results to understand the behavior of the algorithm when the number of heavy atoms is increased. A molecule consisting of 15 heavy atoms is too small. Even small molecules below 500 daltons that satisfy the Rule of 5 are composed of about 30 heavy atoms on average. An experiment showing the percentage of valid molecules generated by the algorithm proposed with more heavy atoms would greatly enhance readers understanding of its effectiveness.

6) Recently, various molecule generation algorithms have been proposed. While these algorithms appear to generate valid molecules in principle, it is unknown if they actually generate synthesizable molecules. Providing information on whether the molecules generated by the proposed algorithms are synthesizable and providing synthetic root analysis results would allow SimGen to generate real, usable molecules rather than insilico-level ones.

7) It is very impressive to control the molecules generated by adjusting the prior, but is it possible to control the properties of the molecules generated by these prior? Or in the case of drugs, the protein in the protein-ligand complex is very important for drug design. Can the binding partner protein's information be considered when generating the molecule? It would be good to answer these questions and, if not, to mention these limitations in the paper.

8) There are a few typos (e.g., line 487 (see Section III A3, the; equation 15, etc.).

(Remarks on code availability)

The code and website provided seems to be well made.

Version 1:

Reviewer comments:

Reviewer #2

(Remarks to the Author)

Thank you for your response to the review comments, but there are some concerns.

1) It was mentioned that performance degradation occurs when the number of reference molecules exceeds 256. However, 256 reference molecules are too few to represent all known molecules. I understand that the methodology proposed in this paper, SiMGen, is intended to be a more general-purpose molecule generation model. However, these experimental results imply that, in order to use SiMGen, one must collect reference molecules suitable for the purpose before the training process and then retrain the model. These questions need to be answered.

2) The response mentioned that no instances of molecules from the reference set were observed in the 4,000 examples. What was the basis for this judgment? More than simply comparing SMILES strings is required for accurate evaluation. A precise comparison through similarity comparison with reference molecules is necessary. Generally, the Tanimoto similarity threshold depends on the type of fingerprints used to represent the molecules [1]. For example, a threshold of 0.7 is commonly used for 1024-bit Morgan fingerprints based on ECFP. A detailed description of the methodology used to calculate similarity more accurately is required.

As an aside, the maximum number of reference molecules is 256, so for a more rigorous evaluation, it would be nice to show that SiMGen can generate novel drug-like molecules by comparing their similarity to FDA-approved drugs published by the drug bank.

3) Considering that the molecules in the QM9 dataset are relatively small, it is understandable that it may be difficult to generate large molecules due to this prior distribution. However, it is necessary to show whether different prior distributions yield results consistent with the authors' arguments, unlike the QM9 dataset. It should be demonstrated whether the rate of generating valid molecules is maintained regardless of the size of the molecules.

4) Similarly for synthesizability, as the proportion of synthesizable molecules tends to decrease as the size of the generated molecules increases. It is necessary to show whether this tends to change when using distributions other than QM9. In particular, there seem to be almost no synthesizable molecules as the number of atoms increases. In such cases, what is the practical significance of this methodology? The limitation is said to be mentioned in section II C3, but it's hard to find it.

[1] Lo, Y. C., & Torres, J. Z. (2016). Chemical similarity networks for drug discovery. *Special Topics in Drug Discovery*, 1, 53-70.

(Remarks on code availability)

The web service that can execute the proposed method works well and appears to be useful.

Reviewer #3

(Remarks to the Author)

While the paper seems to have went through a round of peer-review, this reviewer is requested to write an initial review for the paper: "Zero Shot Molecular Generation via Similarity Kernels".

The authors propose several interesting ideas that are considerably novel compared to existing literature on molecular generation, which are mostly diffusion models. The authors propose to leverage the pretrained representation of an MLIP to define a similarity kernel, from which a denoising vector field can be defined and used for molecular generation. The denoising field needs to be combined with a Particle Swarm Optimisation (PSO) algorithm for sampling the composition of the molecule. The authors demonstrate comparable generative performance to baseline models on a QM9 baseline and several examples on conditional generation with pre-defined priors.

Although this reviewer appreciates the new ideas presented in this paper and find them potentially inspiring for future works, the experimental results and arguments doesn't seem to support the claims on the advantage of the proposed model strongly.

In particular, the authors pointed out three limitations of existing diffusion models:

Lack of scalability. The authors argue the fully-connected graph construction of previous diffusion models make them less scalable and they use a local model. However, this argument by itself does not seem sufficient to establish the scalability of the proposed model. The authors did not present any scalability-related results, such as the runtime for sampling a certain number of molecules. In practice, larger systems also usually use coarse-graining (e.g. the frame representation for protein) to significantly reduce the number of particles in the system. Moreover, according to Fig 5, the generation validity for the proposed model significantly drops with more atoms in the system. For the QM9-like prior, the validity is <40% with 50 atoms, this poses doubts on whether the proposed model is able to perform for larger systems at all, while diffusion models have been shown to scale well (AlphaFold 3, RosettaFold All-atom).

Limited control. This reviewer believes on this one the proposed method allow some interesting conditioning with little effort, but similar things can be done for diffusion models in general. A shape prior / conditional signals from a set of examples is not hard to get incorporated to a diffusion model without further training / finetuning by applying an energy-based guidance

(similar to classifier-based guidance but the classifier is not trained, but pre-defined).

Low transferability. What does transferability mean in this case concretely (e.g., in terms of an experimental metric)? Which experiments demonstrate superior transferrability of the proposed method?

Overall, while the interesting ideas of utilizing pretrained representation for a generative model is appreciated, there are concerns over the scalability and performance of the model. They don't have to beat diffusion models, but current experimental results raise concerns on whether they are generally competitive or can they improve further with more data / better representation etc.

(Remarks on code availability)

code seems satisfactory.

Version 2:

Reviewer comments:

Reviewer #2

(Remarks to the Author)

The authors appear to have adequately addressed the requested inquiries.

(Remarks on code availability)

Reviewer #3

(Remarks to the Author)

I thank the authors for their responses to my comments and questions.

The authors have adjusted their claims and analysis on diffusion-based generative models, and have presented a new perspective regarding their proposed approach, and the claims are now better supported by the experimental results.

However, I still have concerns regarding the scalability of larger systems of the proposed method compared to diffusion models, especially if the proposed model is able to train on and generate larger molecules -- the GEOM dataset [1] would be a great dataset to demonstrate that. What would be the percentage of valid generation? Further, the notion of validity, "Valid atoms are defined as atoms whose number of closest neighbours \leq their natural valence.", is not concretely defined in the main text, and might not be comprehensive. As an example, the tests proposed in PoseBuster [2] could offer a better quantification of generation validity.

[1] Axelrod, Simon, and Rafael Gomez-Bombarelli. "GEOM, energy-annotated molecular conformations for property prediction and molecular generation." *Scientific Data* 9.1 (2022): 185.

[2] Buttenschoen, Martin, Garrett M. Morris, and Charlotte M. Deane. "PoseBusters: AI-based docking methods fail to generate physically valid poses or generalise to novel sequences." *Chemical Science* 15.9 (2024): 3130-3139.

(Remarks on code availability)

code seems satisfactory.

Version 3:

Reviewer comments:

Reviewer #3

(Remarks to the Author)

I thank the authors for their response and revision. The authors pointed out:

> Generation scalability is demonstrated in the "Scaling to larger molecules" subsection (Section II B 4), where we show that our approach maintains a consistently high validity rate per atom (approximately 99%) even for molecules with 50 heavy atoms, which already exceeds the typical size of molecules in the GEOM dataset.

However, Fig 4 B shows poor validity in terms of the generated molecule as a whole. As the authors also pointed out in their response:

> While SiMGen's local approach offers flexibility in incorporating spatial priors of arbitrary shapes, as we will demonstrate in the next section, it also inherently limits its ability to capture the global spatial characteristics of larger molecules.

The above issue in scalability of the proposed approach, to this reviewer, seems like a major limitation. However, the paper does have other merits. In conclusion, I am recommending a "borderline accept" in the sense that if all other reviewers

recommend accepting this paper, I would not argue against acceptance.

(Remarks on code availability)

Response to the reviewers

We thank the reviewers for their time and the constructive criticism. We are encouraged that our approach was found "innovative" (R1) and the ability to control the shape of the generated molecules was deemed "very impressive" (R2).

Both reviewers identified experimental validation to be the weakest part of our initial submission. We have addressed this in two ways:

1. We expanded the evaluation of SiMGen in an unconditional setting, comparing it to other diffusion models and analysing how reference set size and prior shape affect generated molecule quality.
2. We introduced section II C 3, demonstrating SiMGen’s use in designing binders for a supramolecular cage and discussing limitations of local generators as standalone tools for drug design.

These changes clarify how SiMGen compares to other generative approaches and highlight how precise shape control can be used to design molecules for specific binding pockets.

Specific changes include:

- Added Table I comparing SiMGen to trained diffusion models using the metrics suggested by the reviewers, and examining performance across varying reference data sizes.
- Expanded Figure 5 to 50 heavy atoms (previously 15), including comparisons of how prior shape affects valid molecule fraction with increasing atom count.
- Conducted a proof-of-concept experiment using SiMGen to design binders for a toy protein pocket model.
- Adjusted the refinement stage of our approach to improve performance.
- Retrained the hydrogenation model to address a major failing point in our initial version.

We address the reviewers’ concerns point-by-point below.

Reviewer 1

Reviewer Point P 1.1 — Conduct a thorough quantitative analysis using metrics such as validity (how many generated molecules are chemically viable), uniqueness (diversity of the output), and novelty (comparison against known datasets) for unconstrained generation. These metrics are standard in assessing molecular generation models and will provide a solid foundation for evaluating the model’s performance. If possible, compare SiMGen’s performance with other state-of-the-art methods, including diffusion generative models, to clearly show where it stands in relation to existing technologies.

Reply: We have expanded section II B 3 to include a comparison with trained diffusion models. In addition to the suggested metrics, we incorporated Synthetic Accessibility Scores [3] to assess the synthesizability of generated molecules.

Table I shows that SiMGen, using just 256 reference molecules, performs comparably to state-of-the-art diffusion models and excels in novelty. This high novelty is expected, given that SiMGen uses less than 0.5% of the full QM9 dataset, whereas other models typically train on 80% of the data.

Reviewer Point P 1.2 — Demonstrate the model’s capabilities for targeted molecule design in one or two case studies. In my opinion, conditional generation is the most important practical application of models for molecule generation. Case studies could include designing a drug molecule with desired biological activity or desired pharmacophore or materials with particular physical properties. This will greatly enhance the paper’s relevance and applicability, showcasing the practical utility of the model beyond theoretical capabilities.

Reply:

We agree that conditional generation is crucial for practical applications of molecular generation models. To test SiMGen’s capabilities in this context, we chose the supramolecular Octa-acid (OA) system as a case study. OA’s well-defined binding pocket and lack of slow conformational modes make it a useful model system [5].

Using known binders from the SAMPL6 challenge [5], we constructed a point cloud prior for SiMGen. By incorporating this prior and static OA atoms into the generation process, we generated binders with higher docking scores than the original molecules (Figure 8). Notably, the point cloud prior alone prevented clashes with the binding pocket during generation – a problem diffusion-based docking models occasionally struggle with [1].

We describe this experiment in detail in section II C 3. We also highlight some limitations of using a local generator such as SiMGen as a standalone tool for drug design. For designing ligands targeting medically relevant proteins, we envision SiMGen being used in conjunction with other generative models. Combining the scores of a diffusion model trained on drug-like molecules with SiMGen’s spatial arrangement capabilities presents an promising direction for future research.

We discuss how our approach could be changed to target specific physical properties in our response to P 2.7.

Minor

Reviewer Point P 1.3 — Line 283: missing colon after "Low transferability"

Reply: Fixed.

Reviewer 2

Reviewer Point P 2.1 — One of the cores of the proposed algorithm is similarity kernels, which are defined by the RBF kernel. Similarity kernels require reference data, and in this paper, 259 QM9 molecules were used as reference data. The molecules generated are affected by how the reference data is organized; therefore, the authors should analyze the generation model’s results according to the reference data configuration for the readers’ understanding, e.g., how the molecule generation changes according to the reference data configuration, what is the sufficient number of reference data sizes needed for molecule generation, etc.

Reply:

As part of our response to point P 1.1, we included an analysis on how the size of the reference set size affects the quality of generated molecules.

Table I demonstrates that performance improves with increasing kernel size. However, we observed diminishing returns beyond 256 molecules. Considering that computational cost increases linearly with kernel size, we find that a reference set of 128 to 256 molecules offers an optimal balance between performance and efficiency.

Reviewer Point P 2.2 — Also, the proposed score includes a term related to the prior. Therefore, it seems important to add an analysis related to the effect of the prior and similarity energy on the molecules generated. In this paper, there are only results where the molecules generated according to the prior can be controlled. Such an analysis would not only be of interest to the reader but would also make it easier to understand the proposed method.

Reply:

To address the reviewer’s question about the effects of prior on generated molecules, we have expanded our analysis in several ways:

1. As part of addressing point P 2.5, we’ve included an analysis of how the prior affects the overall validity fraction of generated molecules.
2. We’ve added a new case study on the Octa-acid (OA) system, demonstrating how a point cloud prior can be used to design ligands for a binding pocket.

While the similarity energy is the primary driver of the generation process - arranging atoms into locally sensible structures - the prior almost entirely determines the overall shape of the generated molecule. We believe the additional experiment we carried out together with Figures 6 and 7 (already present in the original submission) provide the reader with a clear understanding on the role of the prior in our approach.

Reviewer Point P 2.3 — The proposed algorithm is only compared to a baseline based on linear interpolation. More comparative experiments with various state-of-the-art methodologies are needed.

Reply: We have addressed this in our reply to P 1.1.

Reviewer Point P 2.4 — In the field of molecular generation (e.g., drug generation), one of the key concerns is the novelty of molecules generated by artificial intelligence or machine learning. The proposed algorithm shows very high atom environment similarity to the data used as reference. While this is not a direct measure of the similarity between molecules, it is worth checking that the molecules generated by the proposed algorithm are not only similar to known molecules.

Reply: We’ve addressed the concern about molecular novelty in Table I, which shows that in an unconditional setting, >95% of molecules generated with SiMGen were unique. We observed only one instance across 4k examples where a molecule from the reference set was exactly reproduced.

While high local similarity doesn’t necessarily imply overall molecular similarity, we acknowledge that measuring uniqueness by the fraction of unique canonical SMILES can be misleading, as minor structural changes can result in “unique” molecules.

To provide a more robust assessment of diversity, we calculated Tanimoto similarity distributions between molecules generated by SiMGen (with different reference set sizes) and MiDi [6]. Figure R1 demonstrates that both SiMGen and MiDi generate diverse sets of molecules, with the majority of

Figure R1: Distribution of Tanimoto similarities between molecules generated with SiMGen or MiDi [6], a trained diffusion model.

inter-molecular similarities below 0.2. For context, a Tanimoto similarity of 0.4 is typically used as a threshold when searching for chemically similar compounds [4].

Reviewer Point P 2.5 — The molecules generated by the proposed algorithm show a significant drop in the percentage of valid molecules from around 10 heavy atoms. It seems to be around 0.6-7 when hydrogen is included, but it would be invaluable to see more experimental results to understand the behavior of the algorithm when the number of heavy atoms is increased. A molecule consisting of 15 heavy atoms is too small. Even small molecules below 500 daltons that satisfy the Rule of 5 are composed of about 30 heavy atoms on average. An experiment showing the percentage of valid molecules generated by the algorithm proposed with more heavy atoms would "greatly enhance readers understanding of its effectiveness.

Reply:

We appreciate the reviewer’s insight about the importance of generating larger molecules.

To improve SiMGen’s performance when generating large molecules we made two changes to our generation procedure.

1. We optimised the hyperparameters of our hydrogenation model, and added a single stage of active learning where the model was trained on $\approx 10k$ molecules generated by SiMGen.
2. We noticed that element swapping during the refinement stage using the energies from the pre-trained machine learning force field was in fact detrimental. We disabled it.

With these changes, we reran the size sweep experiment and extended the range of heavy atoms considered in Figure 5 from 15 to 50. Our improvements significantly increased the fraction of valid

molecules for structures with 15 heavy atoms from 0.6-0.7 to ≈ 0.9 . However, we still observe a decrease to ≈ 0.6 for molecules with 30 heavy atoms and further decline for larger structures.

This drop-off is primarily due to the prior fitted to QM9 molecules (a Gaussian with $\Sigma = \text{diag}(1., 1.4, 2.6)$), which forces atoms in the center into high-density arrangements, often resulting in multiple quaternary carbon centers. These configurations frequently cause errors during hydrogenation. Importantly, when we adjust the prior to more open structures (e.g., a flat disc or long line), the fraction of valid molecules increases to 0.6 even for structures with 50 heavy atoms. This demonstrates that SiMGen can generate larger molecules effectively when using an appropriate prior.

Reviewer Point P 2.6 — Recently, various molecule generation algorithms have been proposed. While these algorithms appear to generate valid molecules in principle, it is unknown if they actually generate synthesizable molecules. Providing information on whether the molecules generated by the proposed algorithms are synthesizable and providing synthetic root analysis results would allow SiMGen to generate real, usable molecules rather than insilico-level ones.

Reply: We have added an analysis on the synthesizability of the generated molecules in section II B 3 and Table I.

Our results show that SiMGen compares favorably to other models in the unconditional generation of small molecules (9 heavy atoms) in terms of synthesizability, as measured by the Synthetic Accessibility Score (SAS) [3]. SiMGen generates a higher proportion of easily synthesizable molecules (SAS ≤ 3) compared to both trained models and the reference data.

However, we expect the fraction of synthesizable molecules to decrease significantly as the size of the generated molecules increases. We acknowledge and discuss this limitation in the new section II C 3.

Reviewer Point P 2.7 — It is very impressive to control the molecules generated by adjusting the prior, but is it possible to control the properties of the molecules generated by these prior? Or in the case of drugs, the protein in the protein-ligand complex is very important for drug design. Can the binding partner protein's information be considered when generating the molecule? It would be good to answer these questions and, if not, to mention these limitations in the paper.

Reply:

Our approach allows for several ways to influence generated molecules:

1. Binding partner consideration: As demonstrated in our OA system case study, explicitly including binding pocket atoms in the generation process naturally influences the outcome through similarity energy.
2. Property control: Other properties can be incorporated into the evolutionary part of our algorithm. Recent work by Cremer et al. [2] explored this idea framing it as importance sampling rather than particle swarm optimization.

However, we acknowledge limitations in controlling certain properties, particularly for larger molecules. As SiMGen is a local generator, it may produce incompatible functional groups on opposite sides of large molecules due to the lack of long-range information propagation. For properties like synthesizability, combining SiMGen's score function with that of another diffusion model could be a more effective approach. This limitation and potential solution are now discussed in section II C 3.

Minor

Reviewer Point P 2.8 — There are a few typos (e.g., line 487 (see Section III A3, the; equation 15, etc.).

Reply: Fixed.

References

- [1] Martin Buttenschoen, Garrett M. Morris, and Charlotte M. Deane. “PoseBusters: AI-based Docking Methods Fail to Generate Physically Valid Poses or Generalise to Novel Sequences”. In: *Chemical Science* 15.9 (2024), pp. 3130–3139. ISSN: 2041-6520, 2041-6539. DOI: 10.1039/D3SC04185A. (Visited on 05/22/2024).
- [2] Julian Cremer et al. *PILOT: Equivariant Diffusion for Pocket Conditioned de Novo Ligand Generation with Multi-Objective Guidance via Importance Sampling*. 2024. DOI: 10.48550/arXiv.2405.14925. arXiv: 2405.14925 [cs, q-bio]. (Visited on 05/31/2024).
- [3] Peter Ertl and Ansgar Schuffenhauer. “Estimation of Synthetic Accessibility Score of Drug-like Molecules Based on Molecular Complexity and Fragment Contributions”. In: *Journal of Cheminformatics* 1.1 (2009), p. 8. ISSN: 1758-2946. DOI: 10.1186/1758-2946-1-8. (Visited on 06/24/2024).
- [4] *RDKit Blog - Fingerprint Similarity Thresholds for Database Searches*. <https://greglandrum.github.io/rdkit-blog/posts/2021-05-21-similarity-search-thresholds.html>. 2021. (Visited on 06/29/2024).
- [5] Andrea Rizzi et al. “Overview of the SAMPL6 Host-Guest Binding Affinity Prediction Challenge”. In: *Journal of Computer-Aided Molecular Design* 32.10 (2018), pp. 937–963. ISSN: 1573-4951. DOI: 10.1007/s10822-018-0170-6.
- [6] Clement Vignac et al. *MiDi: Mixed Graph and 3D Denoising Diffusion for Molecule Generation*. 2023. arXiv: 2302.09048 [cs]. (Visited on 11/24/2023).

Response to the Reviewers

We thank the reviewers for their thorough evaluation of our work. We appreciate that Reviewer 3 recognizes the novel ideas in our paper and that Reviewer 2 has practical questions about our approach.

Although we have shown that SiMGen performs similarly to trained models for small molecules, we agree with the reviewers that it struggles with generating large, synthesizable molecules. To address these concerns regarding practical applicability, we have introduced a new section illustrating how SiMGen can guide trained diffusion models. This hybrid approach allows SiMGen to steer generation toward a desired chemical space while relying on a trained model for the global aspects of molecule generation.

We also clarify that SiMGen is not proposed as a drop-in replacement for fully trained diffusion models. Rather, it complements them by expanding the design space and demonstrating that zero-shot generation is possible, particularly in limited-data regimes where only small, specialized datasets are available.

To summarize, we have:

- Reworded the main text to emphasize that SiMGen aims to broaden the exploration of score-based methods rather than compete directly with fully trained models.
- Added a new section demonstrating how SiMGen can guide a trained diffusion model for generating larger, drug-like molecules.
- Included additional similarity checks (Morgan/Tanimoto) in the SI to confirm novelty.
- Made multiple small changes to the text to improve clarity and flow.

We now address each reviewer’s comments in more detail below.

Reviewer 2

Thank you for your response to the review comments, but there are some concerns.

Reviewer Point P 2.1 — It was mentioned that performance degradation occurs when the number of reference molecules exceeds 256. However, 256 reference molecules are too few to represent all known molecules. I understand that the methodology proposed in this paper, SiMGen, is intended to be a more general-purpose molecule generation model. However, these experimental results imply that, in order to use SiMGen, one must collect reference molecules suitable for the purpose before the training process and then retrain the model. These questions need to be answered.

Reply:

We thank the reviewer for raising this concern.

First, in most practical scenarios one does not need to represent all of chemical space; instead, one targets a specific set of chemistries for which data are typically scarce and expensive.

Second, while all generative methods require data collection, SiMGen needs only a relatively small set of reference molecules to perform zero-shot generation. We think that the SiMGen working even with small reference sets (tens of molecules) is in fact a benefit of the method.

Finally, we do not propose SiMGen as a general-purpose generator; rather, it is a demonstration of how a local score-based approach can be used to generate molecules. However, we agree that this point was not clear in the original text and hope that the revised text clarifies this.

Reviewer Point P 2.2 — The response mentioned that no instances of molecules from the reference set were observed in the 4,000 examples. What was the basis for this judgment? More than simply comparing SMILES strings is required for accurate evaluation. A precise comparison through similarity comparison with reference molecules is necessary. Generally, the Tanimoto similarity threshold depends on the type of fingerprints used to represent the molecules [1]. For example, a threshold of 0.7 is commonly used for 1024-bit Morgan fingerprints based on ECFP. A detailed description of the methodology used to calculate similarity more accurately is required.

As an aside, the maximum number of reference molecules is 256, so for a more rigorous evaluation, it would be nice to show that SiMGen can generate novel drug-like molecules by comparing their similarity to FDA-approved drugs published by the drug bank.

Reply:

We thank the reviewer for the helpful suggestion. We now include a more thorough similarity analysis using RDKFingerprints with a Tanimoto threshold of 0.7, as suggested, and update the table accordingly. This analysis is included both in the main text, and with more detail in the supplementary information sections S1 and S3.

Regarding the request to compare with FDA-approved drugs, we agree that evaluating against larger, more complex molecules is worthwhile; however, purely local models will struggle to perceive the more global constraints of drug-like molecules. We make this clear in section II B 4:

While SiMGen’s local approach offers flexibility in incorporating spatial priors of arbitrary shapes, as we will demonstrate in the next section, it also inherently limits its ability to capture the global spatial characteristics of larger molecules. This is a fundamental constraint of purely local models – they lack the capacity to perceive long-range dependencies. Although the fraction of valid molecules generated by SiMGen could potentially be improved by developing more sophisticated methods for sampling valid molecular shapes, a more promising and practically relevant direction lies in exploring hybrid architectures. Such architectures, combining a global component to guide the overall molecular shape with a local component like SiMGen for detailed structure generation, could offer a more robust solution for generating complex and, importantly, synthesizable larger molecules.

That said, we believe SiMGen still has practical utility when combined with a trained model. These are new results presented in Section II D.

Reviewer Point P 2.3 — Considering that the molecules in the QM9 dataset are relatively small, it is understandable that it may be difficult to generate large molecules due to this prior distribution. However, it is necessary to show whether different prior distributions yield results consistent with the authors’ arguments, unlike the QM9 dataset. It should be demonstrated whether the rate of generating valid molecules is maintained regardless of the size of the molecules.

Reply:

Section II B 4 already includes experiments (up to 50 heavy atoms) showing how SiMGen’s zero-shot performance varies with molecular size, and we indeed see a drop-off at larger sizes. We do not expect

drastically different results if the prior distribution were based on bigger molecules because SiMGen operates by combining local atomic environments rather than entire molecules.

Consequently, while we can slightly adjust the distribution of local environments, the method’s fundamental scaling behaviour with respect to molecule size remains similar. To address the broader concern of practical utility for larger molecules, we demonstrate in our new section how SiMGen can guide a trained diffusion model that handles the global structure more effectively.

Reviewer Point P 2.4 — Similarly for synthesizability, as the proportion of synthesizable molecules tends to decrease as the size of the generated molecules increases. It is necessary to show whether this tendency changes when using distributions other than QM9. In particular, there seem to be almost no synthesizable molecules as the number of atoms increases. In such cases, what is the practical significance of this methodology? The limitation is said to be mentioned in section II C3, but it’s hard to find it.

[1] Lo, Y. C., & Torres, J. Z. (2016). Chemical similarity networks for drug discovery. *Special Topics in Drug Discovery*, 1, 53-70.

Reply:

We thank the reviewer for raising this critical question regarding practical applicability. As noted, SiMGen alone struggles to generate large, synthesizable molecules, since it relies on local atomic environments and lacks a global perspective on molecular structure.

To clarify this limitation, we have emphasized in the revised text that SiMGen’s zero-shot generation is best suited for smaller molecules, serving as an exploration of the design space for score-based methods rather than a general-purpose tool. We now highlight these constraints more explicitly in Section II B 4 and in the conclusion as well.

In response to the concern about synthesizability for larger structures, we introduce a new section demonstrating how SiMGen can guide a trained diffusion model. Here, SiMGen provides fuzzy structural guidance to steer generation toward a given chemical space, while the trained model handles the global aspects necessary for synthesizable, drug-like molecules. We believe this hybrid approach offers a more practically relevant application of SiMGen’s local generation framework.

Reviewer Point P 2.5 — (Remarks on code availability) The web service that can execute the proposed method works well and appears to be useful.

Reply: We thank the reviewer for taking the time to try out the tool, and for the positive feedback.

Reviewer 3

While the paper seems to have went through a round of peer-review, this reviewer is requested to write an initial review for the paper: “Zero Shot Molecular Generation via Similarity Kernels”.

Reviewer Point P 3.1 — The authors propose several interesting ideas that are considerably novel compared to existing literature on molecular generation, which are mostly diffusion models. The authors propose to leverage the pretrained representation of an MLIP to define a similarity kernel, from which a denoising vector field can be defined and used for molecular generation. The denoising field needs to be combined with a Particle Swarm Optimisation (PSO) algorithm

for sampling the composition of the molecule. The authors demonstrate comparable generative performance to baseline models on a QM9 baseline and several examples on conditional generation with pre-defined priors.

Reply: We appreciate the reviewer’s accurate summary of our approach. In the revised version, we also introduce a new section demonstrating how SiMGen can guide trained diffusion models, highlighting its utility for steering generation while relying on a learned global prior.

Although this reviewer appreciates the new ideas presented in this paper and find them potentially inspiring for future works, the experimental results and arguments doesn’t seem to support the claims on the advantage of the proposed model strongly.

In particular, the authors pointed out three limitations of existing diffusion models:

Reviewer Point P 3.2 — Lack of scalability. The authors argue the fully-connected graph construction of previous diffusion models make them less scalable and they use a local model. However, this argument by itself does not seem sufficient to establish the scalability of the proposed model. The authors did not present any scalability-related results, such as the runtime for sampling a certain number of molecules. In practice, larger systems also usually use coarse-graining (e.g. the frame representation for protein) to significantly reduce the number of particles in the system. Moreover, according to Fig 5, the generation validity for the proposed model significantly drops with more atoms in the system. For the QM9-like prior, the validity is $< 40\%$ with 50 atoms, this poses doubts on whether the proposed model is able to perform for larger systems at all, while diffusion models have been shown to scale well (AlphaFold 3, RosettaFold All-atom).

Reply:

Although fully-connected graph constructions are fundamentally less scalable than local models simply due to the number of edges, we agree with the reviewer that this limitation is ultimately theoretical. “The bitter lesson” indeed suggests that, given enough compute, these concerns may not be as pressing in practice.

We have revised Section II B to clarify that we view SiMGen as an expansion of the design space for score-based molecular generation rather than a drop in replacement.

Reviewer Point P 3.3 — Limited control. This reviewer believes on this one the proposed method allow some interesting conditioning with little effort, but similar things can be done for diffusion models in general. A shape prior / conditional signals from a set of examples is not hard to get incorporated to a diffusion model without further training / finetuning by applying an energy-based guidance (similar to classifier-based guidance but the classifier is not trained, but pre-defined).

Reply:

We appreciate the reviewer’s positive feedback on incorporating conditional constraints into SiMGen. We agree that shape priors can be readily added via energy-based guidance.

However, designing a well-behaved energy function for more complex constraints can be challenging. In the revised text, we have added Section II D, where we show how SiMGen can serve as a “fuzzy” structural constraint, enabling targeted exploration of specific chemical spaces. We believe this is a novel way to condition trained score-based models using rich many-body information.

Reviewer Point P 3.4 — Low transferability. What does transferability mean in this case concretely (e.g., in terms of an experimental metric)? Which experiments demonstrate superior transferrability of the proposed method?

Reply:

We thank the reviewer for asking for clarification. By “transferability,” we refer to the ability to adapt to new chemical spaces without retraining the generative model.

Since SiMGen relies on local atomic environments and uses a pretrained MACE model for representations, it can operate on a fresh set of reference molecules directly, avoiding the hundreds of GPU hours typically required to retrain a diffusion model. Of course, this assumes that the MACE model performs well on the new chemical domain, which holds reasonably for organic molecules but may break down for more diverse or inorganic systems.

Nevertheless, we find that SiMGen’s inherent “plug-and-play” transferability is particularly beneficial when guiding a trained diffusion model with only a small number of reference molecules.

Reviewer Point P 3.5 — Overall, while the interesting ideas of utilizing pretrained representation for a generative model is appreciated, there are concerns over the scalability and performance of the model. They don’t have to beat diffusion models, but current experimental results raise concerns on whether they are generally competitive or can they improve further with more data / better representation etc.

Reply:

We acknowledge that SiMGen alone cannot outperform fully trained diffusion models in all respects. However, our main objective is to expand the toolkit of score-based methods by introducing a new, local approach rather than supplanting established techniques. In the revised text, we further propose that combining local and global methods may open new directions for molecular generation.

We have also clarified SiMGen’s exploratory role and introduced a new section demonstrating how SiMGen can guide a diffusion model. We believe this hybrid approach holds practical significance, enabling targeted chemical exploration without requiring a complete retraining of a diffusion model.

Reviewer Point P 3.6 — (Remarks on code availability) code seems satisfactory.

Reply: We thank the reviewer for their feedback and are glad they find the code satisfactory.

Response to the Reviewers

We thank the reviewers for their time in evaluating our work.

Reviewer 3 has helpfully identified areas where further clarity regarding the scalability of SiMGen is needed. In response to these comments, we have made targeted revisions to the manuscript to explicitly articulate the zero-shot nature of our approach and clarify why training scalability concerns are not applicable. These revisions aim to ensure a more comprehensive understanding of SiMGen’s unique characteristics.

We give a more detailed response to reviewers’ comments below.

Reviewer 2

The authors appear to have adequately addressed the requested inquiries.

Reply: We thank the reviewer for their comments during previous revisions, which have helped to strengthen our work.

Reviewer 3

I thank the authors for their responses to my comments and questions.

The authors have adjusted their claims and analysis on diffusion-based generative models, and have presented a new perspective regarding their proposed approach, and the claims are now better supported by the experimental results.

However, I still have concerns regarding the scalability of larger systems of the proposed method compared to diffusion models, especially if the proposed model is able to train on and generate larger molecules – the GEOM dataset [1] would be a great dataset to demonstrate that. What would be the percentage of valid generation? Further, the notion of validity, "Valid atoms are defined as atoms whose number of closest neighbours \leq their natural valence.", is not concretely defined in the main text, and might not be comprehensive. As an example, the tests proposed in PoseBuster [2] could offer a better quantification of generation validity.

[1] Axelrod, Simon, and Rafael Gomez-Bombarelli. "GEOM, energy-annotated molecular conformations for property prediction and molecular generation." *Scientific Data* 9.1 (2022): 185.

[2] Buttenschoen, Martin, Garrett M. Morris, and Charlotte M. Deane. "PoseBusters: AI-based docking methods fail to generate physically valid poses or generalise to novel sequences." *Chemical Science* 15.9 (2024): 3130-3139.

(Remarks on code availability):
code seems satisfactory.

Reply:

We thank the reviewer for their comment and agree that scaling is an important concern for generative models. However, we would like to clarify some key aspects of our method regarding scaling.

As emphasized in our title, "Zero-Shot Molecular Generation via Similarity Kernels," our approach is fundamentally zero-shot — it operates without a training phase and relies solely on pre-computed similarity kernels. The kernels are built on **local** atomic environments, meaning that it makes little difference whether these features come from large or small molecules — **locally** they behave identically.

Training scalability concerns therefore don't apply to SiMGen as there is no training process to scale. Such concerns would be relevant for traditional diffusion models, where molecules are typically considered as fully-connected graphs and the size of the generated molecules represents a real distributional shift. We have adjusted Section II B 1 to make this explicit:

$E_{\text{sim}}(\mathbf{x}; t)$ quantifies the similarity of atomic environments in \mathbf{x} to the reference set \mathcal{D}_{ref} – the more similar the environments, the lower the energy. The force on an individual atom can be understood as the direction that maximises the local similarity to the reference data. This similarity force generalises the work of Cobelli et al., allowing the generation of environments that are similar *but distinct* from the reference data. *Crucially, since we are using local descriptors, the approach is independent of the global size of the molecules, from which these environments are taken. The descriptors for each atom are mainly determined by its closest neighbours, meaning it makes little difference whether the environments come from a large molecule, or a small one. This contrasts globally connected generative models where the size of the underlying molecular graph can represent a real distributional shift for the model during training and inference.*

Generation scalability is demonstrated in the “Scaling to larger molecules” subsection (Section II B 4), where we show that our approach maintains a consistently high validity rate *per atom* (approximately 99%) even for molecules with 50 heavy atoms, which already exceeds the typical size of molecules in the GEOM dataset. We already acknowledge that SiMGen’s locality imposes some trade-offs for generating larger molecules in Section II B 4:

While SiMGen’s local approach offers flexibility in incorporating spatial priors of arbitrary shapes, as we will demonstrate in the next section, it also inherently limits its ability to capture the global spatial characteristics of larger molecules. *This is a fundamental constraint of purely local models – they lack the capacity to perceive long-range dependencies. Although the fraction of valid molecules generated by SiMGen could potentially be improved by developing more sophisticated methods for sampling valid molecular shapes, a more promising and practically relevant direction lies in exploring hybrid architectures.* Such architectures, combining a global component to guide the overall molecular shape with a local component like SiMGen for detailed structure generation, could offer a more robust solution for generating complex and, importantly, synthesizable larger molecules.

Indeed we have explored one such hybrid approach mentioned in the above text in section “Combining SiMGen with Trained Models” (Section II D) added in the previous revision. This combined approach is effective for drug-like molecules and enables easy fragment-biased generation. Further exploration of combining local and global models for molecular generation is certainly an exciting avenue of research, but is beyond the scope of the current manuscript.

Finally, we thank the reviewer for pointing out that that atom validity was not fully defined. We have added Section S2 giving a clear definition.

Response to the Reviewers

We thank the reviewers for their time and comments throughout the whole revision process. We appreciate the Reviewer 3’s conditional recommendation of our paper, despite maintaining some concerns about the scalability of SiMGen.

We agree with Reviewer 3 that SiMGen, as a purely local method, shows limitations in molecule-level validity when scaling to larger molecules (beyond ≈ 20 heavy atoms). As the reviewer notes, these limitations are explicitly discussed and acknowledged in the manuscript (Section II B 4):

”While SiMGen’s local approach offers flexibility in incorporating spatial priors of arbitrary shapes... it also inherently limits its ability to capture the global spatial characteristics of larger molecules. This is a fundamental constraint of purely local models...”

The core contribution of this work is the exploration of this local, zero-shot paradigm, demonstrating its unique characteristics: effective generation for smaller molecules without training, fine-grained shape control via priors (Section II C), and the ability to guide existing global models (Section II D). The observed decrease in molecule-level validity for larger, unguided generations is an inherent trade-off for these benefits, stemming directly from the fact we do not model any global properties.

Furthermore, Figure 4B illustrates that even within this local framework, control is possible. Switching from a QM9-derived spherical prior to a linear or disc shape boosts the validity fraction for 50-atom structures from $< 40\%$ to $\approx 60\%$. While optimizing priors could offer further gains, we believe, as stated in the manuscript, that the most promising direction lies in leveraging SiMGen’s strengths within hybrid approaches. This is demonstrated in Section II D (“Combining SiMGen with Trained Models”) and summarized in Section II B 4, where we envision SiMGen ”not as a standalone solution [for generating arbitrarily large molecules], but as a powerful component potentially integrated into hybrid architectures...”

We believe the manuscript accurately presents SiMGen’s capabilities and appropriately acknowledges both its limitations and its place within the field of molecular generation.

Reviewer 3

I thank the authors for their response and revision. The authors pointed out:

Generation scalability is demonstrated in the “Scaling to larger molecules” subsection (Section II B 4), where we show that our approach maintains a consistently high validity rate per atom (approximately 99%) even for molecules with 50 heavy atoms, which already exceeds the typical size of molecules in the GEOM dataset.

However, Fig 4 B shows poor validity in terms of the generated molecule as a whole. As the authors also pointed out in their response:

While SiMGen’s local approach offers flexibility in incorporating spatial priors of arbitrary shapes, as we will demonstrate in the next section, it also inherently limits its ability to capture the global spatial characteristics of larger molecules.

The above issue in scalability of the proposed approach, to this reviewer, seems like a major limitation. However, the paper does have other merits. In conclusion, I am recommending a "borderline accept" in the sense that if all other reviewers recommend accepting this paper, I would not argue against acceptance.

(Remarks on code availability)